# Multi-Sensor HR Mass Data Models toward Multi-Temporal-Layered Digital Twins: Maintenance, Design and XR Informed Tour of the Multi-Stratified Appian Way (PAAA)

**DOI:** 10.3390/s23208556

**Published:** 2023-10-18

**Authors:** Raffaella Brumana, Simone Quilici, Luigi Oliva, Mattia Previtali, Marzia Gabriele, Chiara Stanga

**Affiliations:** 1Department of Architecture, Built Environment and Construction Engineering, Politecnico di Milano, dABClab GIcarus, Via Ponzio 31, 20133 Milan, Italy; raffaella.brumana@polimi.it (R.B.); mattia.previtali@polimi.it (M.P.); marzia.gabriele@polimi.it (M.G.); 2Parco Archeologico dell’Appia Antica, MiC—Ministero della Cultura, Piazza delle Finanze 1, 00185 Roma, Italy; simone.quilici@cultura.gov.it (S.Q.); luigi.oliva@cultura.gov.it (L.O.)

**Keywords:** Digital Twin, Appian Way, multi-stratified, 3D landscape, cultural routes, infrastructure archaeology, archaeological preservation, spherical cameras, Mobile Mapping System, eXtended reality

## Abstract

The article provides an overview of the digitisation project conducted by the Parco Archeologico dell’Appia Antica (PAAA) in Rome, focusing on an 11.7 km section of the Appian Way. This effort is part of the “Appia Regina Viarum” project, supporting the UNESCO heritage site candidacy of the Appian Way. Advanced sensor technologies, including the Mobile Mapping System (MMS), 360° Cameras, Terrestrial Laser Scanner (TLS), digital cameras, and drones, are employed to collect extensive data sets. The primary goal is to create highly accurate three-dimensional (3D) models for knowledge enhancement, conservation, and communication purposes. Innovative tools are introduced to manage High Resolution 3D textured models, improving maintenance, management, and design processes over traditional CAD methods. The project aims to develop multi-temporal Digital Twins integrated with historical documentation, such as Piranesi’s imaginary views and architect Canina’s monument reconstructions. These informative models function as nodes within the DT, serving the PAAA’s geographic hub by means of an eXtended Reality (XR) platform: the paper proposes bridging the physical object and virtual models, contributing to supporting the operators in the maintenance planning as well as information dissemination and public awareness, offering an immersive experience beyond conventional reality.

## 1. Introduction

The uniqueness of the Appian Way stems from its rich and multi-layered historical significance as the “Regina Viarum”—a military road connecting Rome to Brindisi. Throughout its millennia, this road has been marked by its importance as a commercial route adorned with monuments, villas, and sepulchres, serving as a testament to its significance. The Appian Way has undergone various cycles of restoration, transformation, abandonment, and institutional valorisation, reflecting a complex tapestry of events ranging from past magnificence to speculation and preservation. Its journey culminated in the establishment of the Parco Archeologico dell’Appia Antica (PAAA), the largest urban park in Europe, which presents an exceptional opportunity for preserving, showcasing, and implementing sustainable interventions within the context of the ongoing UNESCO candidacy. The digitisation endeavour aims to unravel the Appian Way’s multi-layered identity, where objects and scales intricately interweave while respecting one another. Firstly, the focus lies on the infrastructure itself, examining its construction techniques and material traces. Secondly, attention is given to the landscape, which represents the legacy of the 19th century. Over time, the road has come to be recognised as a unique nature-built landscape, deserving preservation due to its tangible and intangible values that contribute to human well-being. Lastly, the layers of architecture and archaeology are explored, encompassing sepulchres, immovable structures, movable objects, and fragile remnants that require careful classification and analysis.

This paper investigates the integration of massive digitisation requirements, quantitatively addressing the extensive and multi-layered content along the 11.7 kilometres of the Appian Way. The study focuses on various aspects, including the environment and landscape, architecture and archaeology, and movable objects. The aim is to ensure both the quantity and quality of results, particularly in terms of accuracy, encompassing surveying, data processing, informative Atlas creation, modelling, and communication through technologies such as Scan-to-OBJ-GIS-HBIM-to-XR.

Section 2 outlines the Materials and Methods, addressing the surveying objectives of the Appian Way and the coordination activities under the Parco Archeologico dell’Appia Antica (PAAA). It considers the historical uniqueness, maintenance, design, communication aspects, and institutional coordination. Section 3 focuses on the multi-stratified requirements, examining the richness and interconnected values of infrastructure components at different scales, including the road, architectural and archaeological monuments, and the landscape.

Section 4 discusses the State of the Art of multi-sensor data acquisition. Section 5 describes the surveying innovation and presents the accuracy results obtained by comparing different technologies and their combined outcomes.

Section 6 delves into the High-Resolution Digital Twin, precisely the results achieved regarding textured HR 3D models that meet the requirements of the PAAA. It proposes innovative tools for managing the achieved richness, with CAD serving as an output derived from the 3D textured models generated by the photogrammetric process, Terrestrial Laser Scanning (TLS) point clouds, and mobile mapping system (MMS) point clouds. The section also discusses the 3D representation of the multi-scale and multi-layered Appian Way, incorporating vertical and longitudinal sections. Vocabularies are conceived to feed up the linked information on Materials and Construction Techniques linked to GIS and HBIM.

Section 7 highlights the specificity of the innovative concept of the 3D landscape as an additional layer in surveying the Appian Way. It presents the results obtained from photogrammetric data acquisition and optimisation processing.

Section 8 discusses and introduces the Digital Twin concept undertaken so far by the PAAA and future developments. The PAAA 3D models represent a sort of zero level that is progressively projected in a temporal transition: in the current meaning, DT integrates Earth Observation for environmental scale monitoring. However, the PAAA DT concept aims to go one step ahead. On one side, the Appian Way DT includes the Earth Observation monitoring supporting the connection among the physical and virtual model at the geographic scale: even if at the beginning, it represents a support to the landscape preservation of the uniqueness and specificity of the Agro Romano transformation under climate and anthropic threats. Enriching the Digital Twin by including historical maps, perspective views, and photographs, progressively updatable in the future, is crucial to better support a progressive understanding of the multi-temporal transformations that occurred, including the perspective view comparisons facilitated by the 3D textured models. Moreover, the PAAA DT tries to involve 3D advanced textured models in a live DT to be constantly usable and accessible, also to the operators, in a certain way complementary to the BIM and GIS tools.

The case study here presented in the Section of the Marco Servilio Sepulchre can be spanned to all the Appian way sections, each one projected in the past and in the future: the DT based on the 3D immersive models generated aims to demonstrate the potentials of eXtended Reality (VR-AR) web interactive platform for operators, for public and visitors, as a live communication: the platform creates a bridge among the physical and virtual objects to allow users and operators to continuously exchange the information and maps collected, connecting past, present and future data, overcoming complex tools (as HBIM) managing models related to information. The DT is at the beginning, and the platform was delivered on June 2023 in the case of the Villa Quintili at the V° Miles of the Appian Way, progressively hosting the Experimental Section illustrated. The paper proposes the creation of a DT that encompasses metric, semantic, geometric, and radiometric information control for the surveyed section between kilometres 10.7 and 10.8, focusing on the Sepulchre of Marco Servilio. Each element within this DT could be a starting point to communicate the specificities of the Appian Way’s vast and diverse history together with the fluxes of information. Sensors and fluxes of data, including crowdsource able information, could be implemented as further development in the future within the PAAA platform.

The combination of physical and virtual tours provides an opportunity to raise awareness about the uniqueness of the Appian Way and to appreciate the efforts made towards sustainable mobility, preserving cultural heritage as a source of health, boosted by high-quality tour experiences in a fluid hybrid (physical and virtual) multi-temporal space. The eXtended Reality sample incorporates historical documentation, including Giovanni Battista Piranesi’s ‘capricci’ and imaginary views, as well as the monument reconstructions by architect Luigi Canina, who oversaw the Appian Way’s restoration between 1851 and 1855. It also emphasises construction techniques and materials, transferring knowledge often unknown to the general public.

## 2. Materials and Methods: The Appian Way “Regina Viarum” Preservation across the Centuries of a Multi-Stratified Uniqueness Recognised by the UNESCO Candidature

The construction of the Appian Way started towards the end of the 4th century B.C., in 312, to ensure direct communication between Rome and the South of Italy. It takes its name from the censor Appius Claudius Caecus, who started its construction as a consular route of strategic interest. The Appian Way length reported by historical sources is about 360 Roman miles (540 km) from Rome to Brindisi, mostly made with different layers on which large paving stones rested (*Basoli*) [1].

This technique has allowed the preservation of large stretches of the original bottom in perfect functionality to the present day. In particular, the section in question was made on the tongue of the volcanic lava flow of basaltic rock (hence the term *Basolatum*) that extended from the Alban Mountains to the Mausoleum of Cecilia Metella, as witnessed by the Hall of Lava visited at the Mausoleum. Since ancient times the first miles between Rome and the lost Bovillae [2] have focused on the attractiveness of the ‘Regina Viarum’ on which the Roman aristocracy competed for the most prestigious villas and burials as well as temples and shrines of primary importance. The term “Regina Viarum” encapsulates the Appian Way’s distinctive character and diverse range of values. It served as a crucial military route while also functioning as a commercial road. The construction of the infrastructure involved overcoming challenging terrains and implementing measures to reduce travel times, resulting in a unique opportunity to connect with the city while traversing a landscape shaped by natural and human influences. The surveyed section of 11.7 kilometres, stretching from Capo di Bove N. 195 to Frattocchie di Marino, falls within the boundaries of the Archaeological Park of the Appian Way, representing the state-owned property. This area holds exceptional historical significance and has been impacted by numerous deliberate or spontaneous activities throughout the Middle Ages, shaping the disciplines of archaeological, urban, and landscape restoration. From the establishment of the Caetani dynasty (associated with Pope Boniface VIII) to the renowned correspondence of Raphael and Baldassarre Castiglione addressed to Leo X, as well as the era of the Grand Tour, esteemed scholars and influential figures have attested to the Appian Way’s unique significance and received significant attention [3]. At the end of the eighteenth century, Pius VI freed the ancient Appian Way (called Via Appia Antica) from traffic, arranging the Via Appia Nuova as an alternative route. In the first decade of the 19th Antonio Canova, Inspector to Antiquities of Rome, proposed the construction of a park on the site of the consular route from the Capitol to the Alban Hills. In 1820, Pope Pius VII approved the edict of Cardinal Bartolomeo Pacca for the protection of antiquities, ending the destruction and the bare vestiges of the past. In 1852, designed by Luigi Canina, Commissioner for Antiquities in Rome, Pope Pius IX inaugurated the archaeological walk on the ancient Appia up to Frattocchie di Marino, a large open-air museum that, together with the subsequent planting of Cypresses (*Cupressus sempervirens*) and Domestic Pines (*Pinus Pinea*) conducted at the beginning of the 20th century by Antonio Muñoz, inspector of the Royal Superintendence of Monuments, will permanently modify the landscape of the first miles [4]. The post-World War II reconstruction radically changed the structure of the Italian territory and the relationship between city and country, jeopardising the preservation of those values until a few years before they were recognised as a common heritage [5]. In light of the potential loss of a universally acknowledged value, the institutions took measures to safeguard the Appian Way. Firstly 1953, they implemented landscape preservation regulations, recognising the need to protect its unique character. Subsequently, Rome’s General Urban Development Plan 1965 prohibited new constructions across the entire region, designating a significant portion of the initial miles for public green spaces. These actions reflect the authorities’ commitment to preserving the Appian Way’s integrity and ensuring its conservation for future generations. From those years on, the battles of Antonio Cederna and the plan scenarios designed by Italo Insolera and Vittoria Calzolari expanded the theme of the protection of the archaeological context to the urban and territorial dimension linking the preservation of the historical landscape to the preservation of the roman countryside [6].

These instances culminated in the establishment of the Appia Antica Regional Park first (1988) and then the Appia Antica Archaeological Park (2016). The Archaeological Park is an autonomous museum institute of the Ministry of Culture, entrusted with protecting the cultural heritage in its territory of competence and coordinating the enhancement of the Appian Way from Rome to Brindisi. The ministerial interest in the Appian Way was further underlined by the launch of the first candidacy promoted and coordinated directly by the Ministry for inclusion in the UNESCO World Heritage List as a “serial site” of the complete route of the Appian Way from Rome to Brindisi, including the Trajan variant.

### The Appian Way Survey as a Virtual Backbone for Documentation, Conservation, Maintenance, Design, Enhancement, and Fruition

On 23 September 2015, the project Appia Regina Viarum—Valorisation and systemisation of the path along the ancient Roman route was presented in the villa of Capo di Bove on the Appian Wayin Rome FSC 2014–2020, CIPE Resolution 3 of 1 May 2016 (20 million euros divided between MiC and Regions). The aim is to recover the original route of the Appian Way to allow—through “slow” tourist mobility–access and enjoyment of the cultural heritage (historical centres, monuments, landscape areas and archaeological areas) that gravitate on the road [7].

The primary objective revolves around cultural tourism to revitalise the regions affected by the historical Appian Way. Substantial funding has been allocated to peripheral institutes under the Ministry of Cultural Heritage (MIC) to research and develop projects focused on enhancing public archaeological sites. A cognitive approach plays a crucial role in valuing a system that, despite its illustrious reputation spanning two millennia, remains largely unexplored.

The adoption of new survey technologies allows for a comprehensive investigation into the characteristics of well-known and frequently visited areas, such as the segment of the ancient Appian Way situated within the boundaries of the Roman Archaeological Park bearing the same name. These technologies enable a detailed examination of the landscape, infrastructure, and archaeological elements, effectively discretising their properties. Furthermore, they facilitate the extraction of information from the 3D models generated for all identified complexes, which are georeferenced within the geographic space of the Park. Integral to the project is the experimental development of an interactive platform aimed at remotely disseminating the data contained within the Park. This platform employs augmented, mixed, and virtual reality to provide immersive experiences for users. Additionally, a fixed location within the Park allows visitors to explore specific sections of the territory and view famous discoveries from various areas within the Park.

This survey constitutes the virtual backbone to connect other surveys (e.g., georadar, Non-destructive diagnostic techniques, and other ICT processing technologies). The backbone on which to convey information from archaeological excavations and restorations is financed by the National Recovery and Resilience Plan (PNRR) [8] and the National Plan for Investments Complementary to the PNRR [9]; digitalisation texts and historical and current images; development of virtual reconstructions for immersive devices (video mapping, augmented reality, geotags, etc.).

The usefulness of the survey is also linked to defining the basic knowledge framework for the maintenance of the vast heritage of the Park, but also to create services and infrastructure and improve accessibility, to initiate agreements with public and private entities for the recovery of areas and infrastructure, for collaboration in management, for the realisation of events, dissemination through sites and social channels.

The financing of the Appia Regina Viarum project will be used for the rehabilitation of the last miles excluded from that intervention, between the limit of the municipal territory of Rome and the junction with the via Appia Nuova (SS 7) in the town of Frattocchie in the municipality of Marino (RM), going to complete the outdoor museum built by Luigi Canina in the 19th, restoring and adapting it to the current needs. The project, commissioned by the AKA studio comprising architects Federica Caccavale, Alessandro Casadei, and Paolo Pineschi, has leveraged the coordination efforts between the Archaeological Park and the Appia Antica Regional Park, as well as the active involvement of municipalities and local associations. Through established agreements and conventions, the Archaeological Park, operating with institutional, economic, and managerial autonomy, collaboratively manages the abundant cultural heritage and vast green spaces in a participatory manner. This inclusive approach ensures the comprehensive engagement of various stakeholders, enabling the effective preservation and sustainable management of the site’s valuable assets.

## 3. A Multi-Stratified Multi-Temporal Survey of the Appian Way: The Infrastructure, Landscape, Architecture, Archaeology Artefacts and Remains Documentation

The Appian Way represents a complex and multifaceted infrastructure, embodying a convergence of archaeology, restoration, landscape, and cultural routes. It traverses various layers and landscapes, adapting to the topography of its territories. Documenting the interrelationships among the different elements, including infrastructure, landscape, architecture, archaeological sites, artefacts, and remnants, necessitates a comprehensive and multi-scale survey. This paper concentrates explicitly on a specific portion of the Appian Way, namely the section spanning the fourth mile, encompassing kilometres 10.7 to 10.8, as illustrated in Figure 1 and Figure 2.

**A multi-layered infrastructure-archaeology**: the survey aims to document the infrastructure of the Appian Way. Monuments and ruins evoke landscapes that reveal layers of the past. The Appian Way represents a merging of natural and historical features, where the archaeology of the road and the archaeology of the monuments and artefacts intertwine, creating a contemporary landscape [10]. The infrastructure-archaeology layer pertains to the road and the monuments, which benefit from the analysis conducted in building archaeology. Building archaeology analysis involves examining direct and indirect sources [11,12,13], historical and on-site analyses, and studying construction techniques and materials such as the *Basolatum* paved surface, *Crepidines*, *Macére*, monuments, trees, and vegetation. Notably, the road serves as both an infrastructure and an archaeological monument. Furthermore, periodic maintenance is required as tourists and citizens still frequent the road. Therefore geometrical surveys and drawings are necessary to delineate the areas needing restoration work.

**Restoration across the centuries:** the Appian Way is essential for the history of architectural preservation for the many works carried out by artists and architects in the 18-19th century, such as Canova, Valadier and Canina. Their work resulted in an open-air museum where deposits were kept on site by means of the scenography realized with the architectural walls hosting the remains. Furthermore, *Macére* were realised by Canina to identify the border of what was becoming an archaeological ‘boulevard’. The *Macére* were realised with the stones found during the excavation and needed maintenance over the years. Furthermore, elms were planted in the Pontine Marshes during restoration at the end of the 18th century. However, pines substituted elms at the end of the 1930s, which is still the Appian Way’s landscape [14]. Today restoring the Appian Way means preserving the Roman monuments and the restoration works.

**Landscape:** the field of landscape architecture has a long-established tradition of mapping landscape spaces to comprehend their spatial-visual characteristics. This involves manually and digitally creating visualisations to gain insight into the landscape [15]. As digital technology advances, landscape research and design increasingly employ digital visual representations, including virtual reality (VR) environments [16]. When considering the Appian Way, prioritising the construction of an immersive landscape matrix encompassing various elements is of utmost importance. This includes not only the main road but also its monumental structures, as well as the complex vegetation that surrounds it. This matrix establishes and visualises the relationship between spatial patterns and ecological effects. These innovative technologies can sustain landscape development, transformation, and preservation.

**The Appian Way holds intangible and tangible meanings in the archaeological park, cultural routes, and memorial places:** walking through the Appian Way means immersing themselves in a space where archaeology and landscape are intimately correlated. The monuments of the road are witnesses of different historical periods, sometimes challenging to identify, and of the skills of engineers, architects, artisans, and artists who worked on the Appian Way to build the road and the monuments. Intangible know-how is expressed tangibly in the constructions. In the same way, they are witnesses of the intangible value that refers to the history of Rome, from the Romans to their decline till the restoration interventions of the 19th century.

## 4. State-of-the-Art: Survey, Digitisation, 3D Models

Several works in the literature address tangible cultural heritage digitisation and 3D recording [17,18,19,20]. They are generally considering monuments [21], building [22], historical city centres [23], and even rural areas [24]. Fewer works address the historical road [25,26]. Some specificities connected to them pose peculiar issues in the digitisation process. Indeed, historical roads are mainly linear elements extending for several kilometres while their width (roadway and their immediate surroundings) is generally limited to a few tens of meters. In historic roads, the pavement and the roadways need to be surveyed, as well as the different monuments along the road, their connection with modern structures and their surroundings. The geometric documentation of cultural heritage elements can be implemented through various techniques. It is out of the scope of this paper to provide a complete review of them; thus, for some recent considerations, the reader is addressed to the consultation of [27,28]. Here only the main elements will be listed to highlight the specific characteristics of the survey strategy presented in Section 5. Among the available survey techniques, the ones that are more widely used are:

**Global Navigation Satellite Systems (GNSS) and topographic methods:** with those methods, a set of specific points (vertices) and their spatial coordinates can be identified in a local or global reference system [29].

The identified vertices serve multiple purposes within the surveying process. They can be utilized to digitize elements and their geometries, allowing for accurately representing objects through simple shapes. Additionally, these vertices can establish the boundaries of the object being surveyed, aiding in its description. Furthermore, they can function as Ground Control Points (GCPs) for other survey activities, such as laser scanning or photogrammetry.

One of the primary advantages of employing Global Navigation Satellite Systems (GNSS) and the topographic method is the ability to measure the position of identified points across a large area. These methods can provide cartographic coordinates for selected points, enhancing spatial accuracy if necessary. However, when implemented independently, these approaches may not yield sufficient detail in cultural heritage applications. The time required to achieve the desired level of detail may render them economically unfeasible or not cost-effective.

**Scanning techniques:** these methods employ active sensors that directly capture the 3D geometric data of the object through a laser light [30]. These devices retrieve the 3D data by applying different measuring principles, such as flying time or phase shift measurement resulting in the definition of a 3D point cloud of high density. Scanning techniques can be divided into two main categories: static and mobile.

Static scanning involves capturing data using a stationary acquisition system, which typically requires a few minutes to complete. This method has demonstrated its efficacy in documenting complex and irregular surfaces, as well as architectural structures and monuments. However, aligning and registering point clouds obtained from various locations within a given area necessitates the identification of common points or targets and ensuring an adequate overlap between scans. These requirements can limit the efficiency and usability of static scanning in large areas and increase the overall acquisition time.

In contrast, mobile scanning techniques, such as Mobile Mapping Systems (MMS), offer distinct advantages. These systems do not require the acquisition system to remain stationary during data collection. Instead, they allow for the scanning process to occur while the system is in motion. This mobility enables efficient data acquisition across large zones without requiring precise alignment of individual scans. By eliminating the stationarity constraint, MMS facilitates more streamlined and time-efficient scanning procedures. [31]. The system can move, and the path of the instrument can be either measured with a combination of inertial platforms and GNSS systems or by applying Simultaneous Localization And Mapping (SLAM) algorithms [32]. Large MMS are mounted on a car and used for city modelling and road network monitoring. Lighter systems are designed as backpacked or handheld. MMS generally present a lower density of the point cloud and lower accuracy than static laser scanners. When dealing with MMS drift, effects may become significant and specific strategies must be considered [33].

**Image-based and photogrammetry**: these surveying methods are easily portable, and their sensors (digital conventional cameras) have a limited cost [34]. With those methods, it is possible (once the camera is calibrated and images oriented) to easily obtain dense 3D point clouds based on the automatic image correlation. Digital cameras can be applied on different platforms, such as Unmanned Aerial Vehicles (UAV), to reconstruct large areas [35] and for the modelling of terrain and cities. However, significant post-processing of the images to obtain the 3D data that will form the model is necessary, especially if the area to be surveyed is large and several hundreds or thousands of images are acquired. In addition, GCPs are needed to provide georeferencing and avoid deformation during bundle adjustment. An advantage of those methods is the possibility to apply textures and obtain a photorealistic view of the 3D model.

As further discussed in Section 5, this work presents a combination of the previously listed techniques to face multiple resolution requirements for creating the Appian Way Digital Twin.

## 5. Multi-Sensors 3D Survey Techniques

The surveying strategy of the Appian way was shaped according to the final goals of the documentation project. In particular, the main purposes and requirements defined for this project were:-overall documentation (scale 1:100) of the whole Appian way in the area between Capo di Bove and Frattocchie (approximately 12 km) with the buffer area of the sidewall defining the state property and the identification of the main monuments;-a set of 120 detailed cross-sections (scale 1:50) with focus areas in correspondence of the main monument along the Appian way in the area between Capo di Bove and Frattocchie;-a detailed orthophoto (pixel size 5.0 mm) of the paving in the area between Via di Fioranello and Frattocchie (approximately 4 km, 8 km foreseen in the future).-A combination of techniques was defined to target each of them to achieve those diversified goals. In particular:-overall documentation: a survey was designed with a handheld MMS. The choice of an MMS was connected with the high productivity of this survey strategy compared to static laser scanning. The choice of a handled MMS instead of a car-based system is because only the face of the monuments directly facing the road can be surveyed with the latter. Rather, several monuments along the Appian Way do not have a driveway all around them; only walking paths are available. In addition, in the area between Via di Fioranello and Frattocchie, the Appian Way cannot be crossed by cars.-detailed cross-sections: for the cross-section in correspondence with the main monuments or with significant areas, a static laser scanning survey was carried out in the nearby cross-section area. This is motivated by the detailed information requested for producing 1:50 drawings cannot be directly extracted from MMS data. The area surveyed with static laser scanning is limited to a few tens of meters around the position of the cross-section. A limited number of scans (2 to 5) are requested for each of them to keep the acquisition profitable in terms of timing.-orthophoto of the paving: a photogrammetric approach was designed for this task. During the design phase, a UAV survey was considered as a possible choice. However, it was discarded for two sets of reasons: (i) the Appian way in the area between Via di Fioranello and Frattocchie is close (less than 500 m) to the Roma-Ciampino airport, so a UAV survey may determine some dangerous interaction with the airport activities; (ii) the presence of several tall-steam trees (mainly *Pinus pinea* trees). This would make defining a-priori flight plan very complex due to such obstacles. It is worth mentioning that the main aim of the orthophoto is the survey of the pavement, so flighting above the tree crown was not an option. In addition, the requested pixel size for the orthophoto would significantly limit the maximum flight height. For those reasons, a ground acquisition was planned with a device based on masts. At the top of the masts, a non-metric digital single-lens reflex (DSLR) camera was used (Nikon D610—Nikon Corporation, Tokyo, Japan) equipped with a Nikkor fisheye 16 mm lens system (Nikon Corporation, Tokyo, Japan).

The following subsections will describe the different surveys in more detail.

### 5.1. GNSS–RTK

The georeferencing of data acquired from various sensors within a unified framework is of utmost importance in the Appian Way digitization project. Inserting the acquired data into a cartographic framework is crucial, given the extensive coverage of the surveyed area. In this project, the RDN2008 reference system, the Italian standard, was employed for georeferencing purposes. A set of ground control points (GCPs) or targets was utilized for the mobile mapping system (MMS), terrestrial laser scanner (TLS), and photogrammetric surveys. In order to materialize the GCPs, wooden checkerboard targets were placed on both sides of the road. These targets were measured using a GNSS (Global Navigation Satellite System) real-time kinematic (RTK) approach. The positioning system of the Lazio Region provided the RTK correction system utilized for the measurements. The closest permanent GNSS station was located in Rome, with an approximate baseline of 18 km, while the second nearest station was in Ardea, with an approximate baseline of 23 km. The RTK approach allowed for target accuracies on the order of ±2.0 cm in planimetry (horizontal positioning) and ±3.0 cm in altimetry (vertical positioning, specifically ellipsoid height). Conversion between the ellipsoid height and orthometric height was performed by utilizing the undulation of the geoid ITALGEO2005. This conversion accounts for the irregularities in the Earth’s gravitational field, ensuring the alignment of heights measured with respect to the ellipsoid to those referenced to the geoid, representing mean sea level.

### 5.2. Mobile Mapping Survey

For the overall documentation of the Appian Way, the MMS used was the handheld system GeoSLAM ZEB Horizon (GeoSLAM, Nottingham, UK). The instrument has a maximum range of 100 m allowing the survey both of the road and its nearby surroundings, and a resolution of 300.000 points/s while the rangefinder accuracy is (for every single point) between 1.0–3.0 cm. Relative accuracy can be further improved with post-processing operations. The system is based on SLAM technology for scene reconstruction and path estimation. As mentioned, drift effects in MMS data should be prevented with specific strategies. In order to prevent excessive drift, three strategies were adopted: (i) the entire road was divided into sections of an approximate length of 200 m, limiting the maximum acquisition time to 20 min to avoid drifts and computation overload in the SLAM processing phase; (ii) each section was surveyed creating at least one closed loop, i.e., the starting point of the acquisition was also the ending point (Figure 3); and (iii) a set of Control Points (between 7 and 12) were measured along the trajectory in each section to constraint the SLAM processing step.

The Control Point position was measured with the GNSS technique using an RTK acquisition scheme. The measurements were corrected using the data collected by the permanent GNSS stations of the Regional GNSS system, the closest station was located in Rome (ROUN) with an approximate baseline of 10 Km. The same points were also measured during the MMS acquisition by stationing on the points for 10 s. Residuals on GCPs after registration are in the order of ±5.0 cm.

### 5.3. Static Laser Scanning Survey

The area in correspondence with the main monuments was surveyed with a detailed static laser scanning acquisition. The 120 detailed sections were measured, and a limited number of scans (2 to 5) were acquired to obtain the requested level of detail for each section. The survey was conducted with the Terrestrial Laser Scanner (TLS) Faro Focus X130 (FARO Technologies Inc., Lake Mary, FL, USA) covering n. 120 square areas, 50 m × 50 m). For each section, the position of at least three targets was measured in RTK to provide georeferencing of the scans in national cartography (georeferencing accuracy similar to the one obtainable with RTK measurements ±3.0 cm). The Faro Focus TLS clouds have been integrated with the MMS ones and extracted different entities with different accuracies, as illustrated below (Figure 4). The Faro focus areas are addressed to the extraction of the geometric entities (carriage wage, architecture, artefacts) in correspondence with the 110 focus and integrated by the orthophoto (0–4 km); the MMS ones for the crossing infrastructures (as the railways and over passages crossing the Appian Way), landscape-vegetation along all the Appian Way sections.

### 5.4. Photogrammetric Survey: Combining DSLR Camera with 360° Camera

As previously mentioned, a ground-based solution was employed for the photogrammetric acquisition. The photography sessions were conducted using a non-metric DSLR camera, specifically the Nikon D610, equipped with a CMOS sensor boasting 24.3 megapixels (6016 × 4016 pixels) and 35.9 × 24.0 mm dimensions. A fixed focal length fisheye lens, the Nikon 16 mm f/2.8, was selected to facilitate image capture while maintaining adequate overlap. The camera settings were adjusted to an aperture of f/11 and ISO 400, determined through a series of tests to ensure that image quality remained uncompromised even under low-light conditions where the shutter speed was reduced.

In order to meet the pixel size requirements for the final orthophoto, the maximum height for capturing photos was limited to 3.0 m. A mast-based system was devised to achieve this height, comprising carbon fibre masts, a ball head for mounting the camera, and a remote controller for triggering photo capture. Designed to be operated by a single individual, the ball head is configured to capture oblique photographs, allowing the camera to record the ground in front of the operator while excluding their feet. The operational setup spans a distance of 5 m in front of the operator and 10 m in width. Photos are taken with an approximate front baseline of 1.5 m and a side baseline of 5 m.

Ground Control Points (GCPs) were strategically distributed at an approximate spacing of 30 m on both sides of the road and were measured using a GNSS (Global Navigation Satellite System) technique with an RTK (Real-Time Kinematic) acquisition scheme. The overall acquisition yielded approximately 6200 images. The subsequent orientation of the images was performed using Agisoft Metashape. GCPs constituted 70% of the surveyed points, while the remaining 30% served as Check Points (C.P.s) for validation purposes. The final average residual on GCPs amounted to ±2.2 cm, while on C.P.s, it reached ±2.5 cm. These results highlight the accuracy and precision achieved in the photogrammetric surveying process.

360° VIDEO cameras acquisition and frames extraction: a survey with a 360° camera was carried out too (Figure 5). This survey aims to complement the photogrammetric acquisition in areas that are difficult to access, to strengthen the acquisition geometry and to integrate the vegetation layer. The survey was performed with the acquisition of 360° videos (resolution 5.7 k, frame rate 30 Hz) using an Insta360 OneX2 camera.

A set of consecutive frames were extracted from the acquired videos (sampling rate 1 frame per second). Approximately 15,000 360° images were extracted from the video. As previously anticipated, the photogrammetric survey was carried out for the portion of the Appian Way of 4.0 km between Via di Fioranello and Frattocchie.

Finally, a test area (approximately 150 m in length) was set up at the fourth mile of the Appian Way to test the combination of different sensors and platforms and develop a multi-temporal digital twin of the Appian Way. In particular, frame images acquired by a drone platform (DJI Mini 2, SZ DJI Technology Co., Ltd, Shenzhen, China), 360°images acquired on the ground (Insta360 OneX2, Arashi Vision Inc., Shenzhen, China), see Figure 5, and frame images acquired on the ground for the detailed survey of Tomb of Marco Servilio Quarto and the high relief of a male figure. Further details for this test are presented in Section 7.

### 5.5. Multi-Sensors Comparison, Accuracy, Gygabyte, Campaign and Processing

The survey of the Appian Way involved the integration of multiple sensors, each chosen to capitalize on their respective strengths. This approach aimed to leverage the unique capabilities offered by each sensor. For instance, the GNSS survey enabled the establishment of robust Ground Control Points (GCPs) for the Mobile Mapping System (MMS) and photogrammetric survey. The photogrammetric block, on the other hand, facilitated the production of a high-resolution orthophoto of the road’s pavement. At the same time, the MMS data provided valuable information regarding the surrounding context and tree coverage. It is important to note that each sensor involved distinct workflows and exhibited varying levels of accuracy. Table 1 summarises the different surveying strategies employed, considering the data types and outputs generated and the associated accuracies and resource requirements. The table also includes information on the effort required for the acquisition stage, expressed in terms of the number of days and operators involved. This overview provides a comprehensive understanding of the diverse approaches adopted in the survey process, facilitating effective data integration and analysis.

## 6. 3D Representation of the Multi-Scale-Multi-Layered Appian Way: The Infrastructure Component, the Architectural and Archaeological Component, the Landscape Component

### 6.1. Photogrammetric Image Blocks Optimisation Targeted to the Appian Way Layers

The Appian Way surveying required the optimisation of data acquisition, data processing and representation of the multi-stratified document: the infrastructure layer, landscape layer, architectural layer, and archaeological remains. The geometric and material component surveying and documentation represents one of the bases upon which different experts, architects, engineers, geomatics, archaeologists, urban planners and landscapers are used for the analysis, design purposes and interpretation [14].

To obtain textured 3D models able to accurately describe the different layers, the different photogrammetric blocks were processed, optimising the data source and experimenting with the best merging results. In the first massive processing phase, the blocks acquired by the NIKOOR camera, integrated with the frames extracted from the 360° images, were processed all together. Then, in a second phase, the optimisation was better studied according to the morphology of the elements to be represented by removing the portions of noise and noise, as explained in Table 2. Notably: to better represents the *Macére* drywall on the borders and the archaeological and infrastructural details, the HR Ground image data (+UAV ground 1–5 m, where available in the test area) have been separately processed, adding the 360# video frames just in the regions lacked (i.e., for vegetation obstacles) due to the flexibility of acquisition near the ground level (as explained hereafter in Section 6.2); to represent the Vegetation Layer the 360° image blocks with the UAV tests have been coupled to gain a fully immersive 3D landscape scenery (as explained in Section 7).

### 6.2. The Photogrammetric Image Block Processing to Obtain an Accurate Model (Mesh Sizing and Texturing)

To capture the details of the infrastructure layer, including archaeological remains and the *Basolatum* paved surface, in the focus of the Sepulchre by Marco Servilio, a survey was conducted using a combination of UAV and ground cameras. The UAV survey was performed using the DJI Mavic Mini at an altitude of 15 m, following a double grid schema with 80% longitudinal overlap between images along the strip and 60% transversal overlap between different strips. The flight height was selected to obtain a final orthophoto with a 5.0 mm Ground Sampling Distance (GSD). A second UAV flight was conducted to capture the upper part of the monuments and the *Macére* and to integrate the aerial survey with the terrestrial survey. A total of 2182 UAV photos were acquired and processed.

The ground survey was carried out using a non-metric DSLR camera, specifically the Canon EOS 5D Mark IV (Canon, Tokyo, Japan) with a 30.4-megapixel CMOS sensor (4464 × 2976) and dimensions of 26 × 24 mm. The camera had a fixed focal length of 20 mm f/2.8 and was set at f/8 and ISO 1250. A total of 2045 pictures were acquired, following the same overlapping schema used for the UAV survey, and processed accordingly. Initially, the cameras were aligned together to maintain a consistent reference. The dense cloud generated in Agisoft Metashape comprised approximately 42,198,010 points, while the 3D textured mesh model comprised 9,803,910 faces (Figure 6). Subsequently, the cameras of the Sepulchre were selected and processed separately to obtain a more accurate 3D model of the monument. The same process was carried out for the *Basolatum* paved surface (Figure 6).

The Orthomosaic represents just an output to be managed within CAD and GIS tools (Figure 7).

### 6.3. The Immersive 3D Model Representation of the Multi-Stratified Appian Way: A Shift toward 3D-Based Tools: The Outputs Generated from the 3D Model Management

In the phase of identification of the final products, it was immediately evident that the pure 3D CAD or GIS 2.5D management of the photogrammetric result (e.g., orthophoto projected in the different planes, for example, the planimetric surface, DEM) could not be considered as the only complete tools of the complexity of the documentation and the entities to be described. Therefore, the CAD output, as shown below, becomes one of the descriptive intermediate links, although basic in the extraction phase of vertical and horizontal 3D profiles of all entities and the daily work practice of operators.

The core of the output is represented by the 3D textured model: a comprehensive, immersive 3D landscape representation of the Appian Way was achieved by integrating the different 3D models and vegetation processed with 360-degree pictures to be managed by operators and professionals within pure modellers such as McNeel Rhinoceros© (version 7.30) to better support the design of the services on course. Such models have been used by professionals to contextualize the preservation and design project phases (Preliminary, Definitive, Executive Design, November 2022) and the procurement selection of the restoration and construction company. Therefore, in addition to the traditional photogrammetric output, the texturised full 3D model was also obtained (not manageable by CAD) so that it was manageable by pure modellers (e.g., Rhinoceros Mc Neel©) [36]. Although not yet used extensively by operators and designers, this type of output is the most user-friendly for the contextualisation of the project, as in the case of slow mobility. At the same time, it becomes the management vehicle in the metaverse of the interactive navigability of models and related information. It is a real immersive model available to designers and coordination of activities by the PAAA. The immersive 3D model representation of the multi-stratified Appian Way (Figure 8) has been generated to describe the 3D landscape, vegetation, and infrastructure, with the *Basolatum* and *Crepidines detailed 3D models*, architectures, archaeological artefacts and remains.

The 3D Immersive model has been obtained by merging all the 3D data, notably: (i) the 3D entities (as the longitudinal section and the carriage-wage line with discontinuity respect to the *Crepidines*)—the 3D line shows the altimetric ascendent volcanic geomorphology of the Appian Way with an altitude difference of 106.08 m along the 11.7 km; (ii) the 3D entities extracted from the TLS and MMS (as the movable archaeological sites and artefacts); (iii) the 3D Textured Objects models coming from the photogrammetry including the 3D vegetation, where present; (iv) the point clouds to complete the 3D perspectives views to support impact analysis of the design of the services or planning the re-vegetation. Figure 9 shows the 3D textured model (OBJ) in correspondence with the Marco Servilio Sepulchre.

The overall model is the result of the integration of the textured model (OBJ) with the 3D features extracted by the cloud points, as the Carriagewage lines along the 12 km, *Basolatum* Roman paving, the sidewalks with the *Crepidines* and *Macére* remains, the longitudinal and cross sections and the focus of the archeologic area (Figure 9). For each section 100 m in length have been derived the grip-models conceived as boxes where to relate in the Digital Twin the collected information coming from the HBIM by means of the XR platform.

### 6.4. Surveying to Interpret and Manage the Uncertainty and Liquid Borders of Decays and Fragile Remains

The 3D model obtained has made it possible to describe not only the most conserved elements or the best-preserved portions but also the deficient portions with varying degrees of degradation and loss of consistency. The result becomes a working tool for operators for the planning of conservative interventions and emergencies in the areas of most significant risk: it offers better support to the estimation but, above all, a tool of three-dimensional analysis of the tangible conditions to identify the priority of the participations in the fragile contexts. As is the case of the borders of the carriage wage, where we do not have finite discontinuity lines separating the carriage wage from the sidewalks: *Crepidines* are primarily undefined. However, stones are emerging from the ground and need to be preserved. Sometimes they are still readable. Sometimes they have totally disappeared (Figure 10). Sometimes we documented the decay and disruption, as in the case of the Macére (Figure 11).

The result is a precious document of the state of preservation and decay that can be used in maintenance planning. The historical-collected data have to be correlated with the current state of conservation of the monuments. The geometrical survey of the section of the road was carried out with the procedure previously described. However, the dataset was integrated, acquiring terrestrial and UAV pictures for photogrammetry.

Particular attention was given to obtaining the polygonal blocks of the road *Pavimentum* (*Basolatum*), the *Macére* and the trees (*Pinus pinea* and *Cypresses*) at the road level, together with the sepulchres and monuments along the Appian Way. Other specific pictures were taken to map the material decay of the monuments. The geometrical survey test was carried out with a Mavic Mini DJI 2 drone to acquire zenith images of the road. Other pictures were taken to capture the trunk and foliage of the trees (360° video where not allowed for the Drones acquisition). Terrestrial photos were acquired with a Canon EOS 5D Mark IV along the two tombs, the polygonal blocks of the road and the *Macére*.

### 6.5. From 3D Textured Model to CAD to GIS Geographical Atlas Data Management of the Layers

From the textured model integrated by the point clouds and the 3D generations extracted, it has been obtained n. 3 Portfolio CAD volumes of the Appian Way from Frattocchie to Capo di Bove (0–4 km, 4–8 km, 8–12 km) with a total of 240 panels, as illustrated in Figure 12 and Figure 13. The 11.7 km section has been subdivided into n. 117 panels, 100 m in length each (leaving variable their height correspondent to the carriage width 15–35 m): the format supports the A0+ layout plotting at the scale 1:100 and n. 3 A3 format portable Portfolios volume). A grid section schema guides users into extracting the different portions, with the progressive kilometre of each section panel also referred to as the transversal sections (bottom). UTM and geographic coordinates are inserted to allow fast Google-based on-site visits to the geometric portfolio to the PAAA operators.

Here are illustrated the series referring to the Marco Servilio Sepulchre, Section Panel n. 108 (from 010.7 km to 010.8 km) with the two adjacent panels 107 and 109: according to PAAA, to support further analysis, for each portion it has been generated 3 different layouts, precisely: ‘0108′, geometric CAD restitution; ‘0108a’, the environmental survey with the clouds coming from TLS, MMS (and Photogrammetry) complementary to the CAD drawing to support the 2D design projects; ‘0108b’, the Orthophoto test details (so far the orthophoto has been funded just for the 0–4 km section). Future campaigns are on planning. Each section layout is integrated by the transversal section (here, the n. 85 surveyed in correspondence of 120 focuses on the altimetric quotation (Figure 12).

The longitudinal section has been integrated with the view fronts section along the 11.7 km with the *Macére*’ walls in red and the vegetation fronts, the 100 m planimetric views and details (Figure 13).

All the entities digitised in the 1:50 scale have been managed as closed self-consistent polygons inherited from the cartographic specifications. This step made it possible to manage the entities as Geopackages layers within the GIS/WEBGIS tools (Figure 14): GIS data management would support the PAAA for maintenance, design, and preservation and the metric-cost computation during the public procurement tenders as in the case of the green maintenance. The metric computation result is summarised hereafter and in the GIS Attribute Table in Figure 14.

**# Sections of ‘Pavimentum’**, paved surface (‘*Basolato*’) still visible (Tot. N. 122 entities, 6400 mq);

**# Carriage paved Pave’** (Tot n. 165 portions, 30,000 mq AREA);

**# “Margines” or “*Crepidines*”:** the two lateral bands up to the state domain borders, ~14 m from the carriage side. Hewn stones stuck vertically into the ground to contain the lateral thrust (more than n. 1900 portions classified, 3400 mq AREA);

**# *Macére*:** state-domain Appian Way border dry walls (Tot n. 1200 portions, 56,700 mq AREA);

# **Archaeological artefact, Unmovable objects,** Tot n. 325 entities, 15,000 mq AREA;

**# Archaeological artefact, Movable objects:** Scattered stone/marble elements: Tot n. 2436 entities, 1575 mq AREA.

**# Green areas to be mowed** (ordinary maintenance and extraordinary maintenance to free the green areas from brambles that overcame the carriage and sidewalk): 221,769.5 mq AREA; and ~brambles 29,388 mq AREA.

**# Tall trees:** Domestic Pine (*Pinus Pinea*) and *Cypress* (*Cypressus sempervirens*): Tot N. 569 entities.

It is the case of the mowing areas or the management of archaeological units, as the still preserved *Basolatum* Roman paving (Figure 14, Upper). Object-based restitution is the basis for object modelling, as illustrated hereafter. It will be the base for the HBIM infrastructure as obtained in the HBIM first output of the Claudius-Anio Novus Aqueduct [37]. An external DataBase (D.B.) conceived as a vocabulary of the Appian Way to be shared among GIS, HBIM, and eXtended Reality for operators and touristic purposes, has been undertaken under the PAAA coordination. It is progressively enriched with the description of materials, construction techniques, and historical construction phases (and restoration phases) (Figure 14, bottom).

Born to manage the common ‘I’ connecting HBIM and GIS, the D.B. has been developed to enrich the personalised parameters (as in the case of the Stratigraphic Units of the Claudius Aqueduct and Cecilia Metella Mausoleum HBIM). The result supports synchronic and diachronic comparisons among different constructions and restoration phases across the centuries in the overall PAAA extension, which will be shared with visitors through the XR platforms [38].

## 7. 3D Landscape Documentation of the Appian Way: Surveying and Representation

### 7.1. Survey Data, Camera Positions, Image Alignment, and Level of Accuracy

The workflow in Metashape for building the vegetation of the Marco Servilio Sepulchre section involved the extraction of 1437 frames from the 360° survey video, with a resolution of 5760 × 2880, and 460 frames from the drone survey, with a resolution of 4000 × 3000. During the investigation, parameters related to dense cloud generation, such as “Quality” and depth maps filtering, were kept constant to ensure consistent results. Other processing parameters, such as image alignment and ground control points/accuracy, were also held constant. Direct georeferencing for UAV images improved the image alignment process by allowing intelligent comparisons between nearby images, thereby increasing computational efficiency and the likelihood of successful image alignment, proving to be particularly beneficial in texturally complex vegetated scenes [39]. The alignment of the photos and, consequentially, the whole workflow was carried out in Agisoft Metashape SfM Software (version 1.8.0). In order to optimise the geometric accuracy of the SfM reconstruction, a full photogrammetric bundle adjustment was performed using both direct georeferencing of image data and ground control points (GCPs). Error terms were specified for directly georeferenced camera positions (±15 m) and GCPs (±0.015 m), with specific markers (PT19, PT20, PT22, PT23, PT24, PT25) used for the GCPs. Regarding depth filtering, the Metashape User Manual provided recommendations based on filtering modes such as Mild, Moderate, Aggressive, and Disabled. In this case, mild-depth filtering was chosen to reduce erroneous points in the dense point cloud while retaining points associated with small features of interest, such as vegetation elements. This choice was particularly relevant for forested environments [39]. For computational efficiency and accessibility, the models obtained from the 360° camera survey and the DJI Mini 2 drone survey were processed at a medium resolution. This decision balanced data accuracy and computational resources, ensuring a reasonably detailed representation of the Appian Way and its surroundings while managing processing time and storage requirements. Integrating higher-resolution data from additional cameras improved precision and accuracy, particularly in small-scale archaeological features, enhancing the model’s overall quality. By following this workflow, the resulting 3D model of the Appian Way encompasses a comprehensive overview of the environment, capturing intricate small-scale archaeological features with high-resolution textures.

### 7.2. Pre-Processing Optimisation of the Dense Cloud (360° Images)

Before pre- and post-processing optimisation, the model incorporating the 360° and drone images, captured using the aforementioned parameters, encountered specific issues and disturbances attributed to the 360° survey data images. 360° image orientation and reconstruction with camera positions constrained by GNSS measurements allows operators to merge the drone tests [40]. The limited directional control of 360° cameras posed a particular challenge. As these cameras capture the entire surrounding environment, there is limited control over the specific viewpoint or focus direction during image acquisition. Consequently, it became difficult to highlight specific details or perspectives within the captured data, resulting in disturbances within the final model. While the comprehensive coverage offered by 360° cameras can be advantageous, enabling a holistic representation of the surveyed area and facilitating spatial relationships and immersive visualizations, it necessitates a pre-processing step. In the case of environmental photogrammetry, this involved employing manual editing techniques to mask out the sky in each frame. This step was essential to mitigate potential interference caused by the sky in the subsequent modelling and analysis processes. By removing the sky from the images, the focus could be directed towards the relevant terrestrial features and ensure a more accurate representation of the surveyed area.

It is worth noting that despite the challenges associated with 360° survey data, integrating such data into the overall model still yielded valuable benefits, particularly in terms of comprehensive spatial coverage. The subsequent pre-processing techniques addressed specific limitations and disturbances, ultimately enhancing the accuracy and reliability of the final model (Figure 15). This step was essential to ensure that the sky did not interfere with the desired focus and accuracy of the reconstructed model. By addressing these challenges through pre-processing techniques and editing, the resulting model could effectively showcase the environment, providing a comprehensive view from all angles while minimising disturbances caused by the inherent limitations of 360° cameras.

The pre-processing stage, which involved the removal of noises, uncertainties related to clouds, and masking out the sky portion of the 360° frames, introduced additional complexity and time to the data processing workflow. However, this step was essential to ensure accurate and focused analysis of the desired subject within the captured 360° frames. The presence of the sky in the frames hindered the analysis and visualization of the main subject, necessitating its removal. In order to minimize disturbances in the final textured model, a filter was applied to the Drone Dense Cloud. The filtering process encompassed a point cloud spacing of 0.034 m, and points were selected using a mask with an Edge Softness value of 1. This filtering operation effectively enhanced the quality of the Drone Dense Cloud, mitigating unwanted noise or artefacts that may have been present. Following the pre-processing and filtering steps, the resulting 360° Dense Cloud, with the disturbances and sky coverage removed, comprised 33,557,803 points. This dense cloud was generated using a medium-quality setting and mild filtering. The Metashape interface visually depicted the 360° survey acquisition, comprehensively representing the captured data for subsequent analysis and interpretation (Figure 16).

### 7.3. Pre- Processing Optimisation of the Dense Cloud (Drone Images)

A filter was also applied to the Drone Dense Cloud to reduce disturbances in the final textured model further. The filter adjusted the point cloud spacing to 0.003 m and applied involved the Compact, dense cloud function. This filtering process helped improve the quality of the Drone Dense Cloud by reducing noise and optimising the point distribution. After filtering, the Drone Dense Cloud underwent a manual cleaning process to address any remaining significant disturbances, particularly errors in the vegetation structure. These disturbances were carefully identified and corrected to enhance the model’s overall quality. The pre-processed Drone Dense Cloud, with the disturbances removed and the sky coverage masked out, resulted in a Raw Dense Cloud consisting of 33,180,126 points (Figure 17).

### 7.4. From Dense Cloud to Textured 3D Model

The final Dense Cloud achieved higher precision and reduced disturbances by implementing these steps, allowing for a more accurate representation of the surveyed environment in the forthcoming textured model. The final Dense Cloud product is obtained through (i) the pre-processed Dense Cloud generated from the 360° images and (ii) the pre-processed Dense Cloud generated from the drone images. These two Dense Clouds were aligned and merged (with Point Cloud assets settings) to create a unified representation of the surveyed area. The resulting final Dense Cloud consisted of 52,448,310 points, and it was generated with a medium-quality setting and mild filtering. This configuration ensured a balance between maintaining a reasonable level of accuracy and managing computational resources efficiently. The file size of the final Dense Cloud was 689.51 MB, reflecting the substantial amount of data captured and processed throughout the workflow (Figure 18).

### 7.5. The Value of Environmental-Landscape Scale Immersive 3D Reconstruction

The comprehensive 3D environmental model with realistic textures that accurately document the multiple vegetational layers of the Appian Way assessed the integration of various sensors and platforms while simultaneously developing a multi-temporal digital representation of the Appian Way. The utilisation of a 360° camera survey played a pivotal role in reconstructing an immersive model representing the environment of the Appian Way, with a primary focus on its application within the realm of environmental photogrammetry. By capturing the surroundings from all angles, the 360° camera survey facilitated the acquisition of a comprehensive dataset, creating a highly realistic and interactive virtual representation of the vegetational matrix. This immersive model allowed for meticulous exploration of the road and its surrounding landscape, presenting a distinctive perspective for environmental photogrammetry applications. Integrating the DJI Mini 2 drone survey further augmented the environmental photogrammetry capabilities of the digital twin model. The aerial imagery captured by the drone introduced an additional layer of information, facilitating the identification of larger-scale environmental patterns, topographical features, and land use characteristics. Through the amalgamation of data from both the 360° camera survey and the drone survey, a comprehensive and multi-dimensional understanding of the Appian Way’s environment could be achieved. The resulting multiscale immersive 3D model, derived from the 360° camera survey and complemented by integrating both the drone and camera surveys, provides a robust platform for advancing environmental photogrammetry. The final product is an innovative DT tool for studying and monitoring the environmental aspects of the Appian Way, contributing to a deeper comprehension of its historical context, where final users and experts can study vegetation patterns, assess land use changes, and perform environmental assessment and monitoring (i.e., carbon capture indicators), thus, ensuring landscape heritage preservation for future generations (Figure 19).

## 8. The Appian Way Multi-Temporal Digital Twin and Future Developments

A further discussion element concerns the definition of the Digital Twin (DT) of the Appian way. In the literature, different definitions of Digital Twin exist. A consolidated and generalized definition for a Digital Twin is “a virtual representation of a physical system (and its associated environment and processes) that is updated through the exchange of information between the physical and virtual systems” [41]. This exchange of data between physical and virtual systems in different application domains (e.g., facility management, transportation sector, manufacturing, etc.) is carried out through real-time data exchange of information based on sensors and IoT systems. However, in the Cultural Heritage field, real-time communication between physical and virtual entities is generally non a major requirement (exception made for real-time monitoring of climatic conditions [42]. In the Cultural Heritage field, the availability of the model to store information coming from the physical world, its ability to store information from different data sources and the possibility to structure them in a spatially organized way seems to be a more important aspect of a Digital Twin [43,44]). In the same way, a major requirement is the possibility of having a structured database to query the system and extract useful information to be used for further simulation and actions. Indeed, the frequency of data updates in the DT for CH can be quite low, some information can be updated daily or weekly, but more often, they are updated seasonally (e.g., state of conservation and decays of materials) or yearly. For some information, no sensor system exists to asses them, and human evaluation is still a key element in the process. In this sense, the system developed for the Appian Way can be defined as a DT since it allows PAAA operators to continuously update the system and add new information, creating this way the exchange of data between physical and virtual systems.

But, the PAAA DT has the ambition to go one step ahead toward the easy accessibility of complex models for the operators, curators of exhibitions and visitors.

In recent years, there has been a significant proliferation of 3D modelling tools and Building Information Modeling (BIM) projects, leading to a transformative impact on the design process. These advancements have enabled designers to seamlessly import and trace their decisions, facilitating a smooth transition from the conceptual stage to a digital model. By employing three-dimensional sketching and modelling techniques, professionals can gain a comprehensive understanding of the intricate form, volume, and proportions of objects or spaces, extending this process to encompass historical buildings and contemporary heritage. Consequently, this approach enhances these structures’ interpretation and diachronic comprehension, providing a tangible and realistic representation of the artefact and its design concept. This, in turn, fosters an enhanced spatial understanding of the historical phases, thereby facilitating improved communication during the final design and construction phases.

On the other hand, static digital representations such as textured mesh and Historic Building Information Modeling (HBIM) models lack the interactive capability to respond to user input in a digital environment. This study addresses this limitation by exploring the understanding, study, and application of visual programming language (VPL) in association with digital models. The research defines a process to create a new digital platform characterised by interactive virtual objects (IVO), where Virtual Reality (VR) becomes a genuine digital twin (DT) capable of supporting various types of users through cloud and web-VR interfaces, ranging from professionals to virtual tourists.

The multidisciplinary approach employed in this study breathes life into static 3D representations, enhancing the virtual experience and promoting information sharing through a DT characterised by IVOs that can interactively respond to user inputs. Leveraging the capabilities of VPL, physical properties such as weight, gravity, and geometric boundaries are linked, rendering them usable, manipulable, and transformable in the VR environment.

The interoperability of formats and packaging functions allows for the development of models that can be seamlessly synchronised with the immersive environment, eliminating the need for repeated phases of model export, import, and saving. Real-time synchronisation, a typical feature of an actual DT, enables simultaneous work in the modelling and cloud environments, resulting in significant time savings during virtual scene setup.

The concept of digital twins originated in the field of computer-aided design (CAD) and manufacturing, where it initially referred to a virtual representation of a physical object or system. However, in the context of built heritage, the term “digital twin” encompasses more than just a 3D model. It involves the capturing, storing, and monitoring of diverse information about the object of interest in a spatial and semantic database. This broader interpretation recognises the importance of visualising and simulating the physical object while capturing and organising various associated data points. These data points can encompass attributes such as historical information, material composition, structural properties, maintenance records, and other relevant information. By organising this information in a spatial and semantic database, researchers and practitioners can better understand the different characteristics of the built heritage. This enables them to analyse, monitor, and manage the objects more effectively, considering both their physical representation and associated data.

The concept of a digital twin has emerged as a valuable approach for managing complex environmental contexts characterised by various phenomena and threats during their ongoing development. It involves establishing a connection between physical reality and its virtual representation, enabling the observation and analysis of their behaviour over time. Real-time data captured from the physical entity through sensors and other data sources are utilised to create a virtual platform that can be analysed, monitored, and manipulated. The overarching objective is to comprehend the ongoing transformations by linking and monitoring past events across remote and recent decades and current developments. This virtual platform can be continuously enriched with data streams from various sources, such as IoT sensor networks and satellite observations. These data fluxes contribute to a deeper understanding of the ongoing phenomena, encompassing a suite of tools, data, models, services, and accessibility, thereby enabling the simulation of future scenarios. The utilisation of digital twins becomes a proactive means to prevent, mitigate, adapt to, and enhance resilience against threats, including climate change and human activities. Therefore, it is crucial to consider the specific context in which the term is being used in order to understand its intended meaning accurately.

This research here focuses moves forward the development of various orientations and enrichments of digital twins at three different and related levels:The Appian Park DT was enriched by EO Digital Twins integration and by metric historical data (Section 8.1)The study explored the integration of Earth Observation (EO) data into digital twins, particularly in the case of the Appian Park DT. This integration enriched the digital twin with historical metric data, enhancing its ability to capture and represent changes over time. The DT has been undertaken as illustrated below.Immersive DT enriched by informed models raises awareness and visitors’ well-being (Section 8.2)The study aimed to create immersive models that prioritise visitors’ awareness rising, fostering their well-being in the fruition of physical and virtual spaces thanks to a DT considering emerging paradigms such as interactivity, immersion, and interoperability in virtual reality (VR) and web-VR environments. These models aimed to provide visitors with an engaging and informative experience while ensuring their comfort and satisfaction. Starting from the Villa Quintili web interactive immersive XR DT implemented on the V° Mile [45], an experimental sample is here developed and presented.Digital Twin enriched by non-metric perspective view comparison rising comprehension (Section 8.3)The study explored the enrichment of digital twins by comparing non-metric perspective views. This approach allowed users to access a virtual tour across the centuries, providing a unique and comprehensive understanding of the object or site represented by the digital twin. In the future crowdsourcing fluxes of images (i.e., on the state of conservation) acquired by visitors could be added to the DT.

By focusing on these orientations and enrichments, the study aimed to advance the capabilities and applications of digital twins in various domains, promoting their usefulness in capturing historical and spatial information, enhancing visitor experiences, and facilitating immersive exploration and analysis of objects and spaces.

### 8.1. EO Digital Twins and the Appian Park DT Enriched by Metric Historical Data

Numerous research efforts are currently dedicated to exploring the potential of DT applications in the context of the Green Transition. By harnessing the power of the DT, stakeholders can make informed decisions, develop strategies, and implement measures that foster sustainable development and environmental stewardship [46].

The availability of Big Data cubes (as the satellite and IoT) in Earth systems science can be fostered in the phenomena understanding thanks to Deep Learning and Artificial Intelligence toward DT [47].

In the Earth Observation field, the possibility to manage DT to monitor desertification effects, land degradation and regeneration actions (as regenerative agriculture) thanks to the availability of historical, current and future, coupled with ground local data (i.e., Drones, sensors, T°, raining data) are allowing the operators, PA, farmers, and associations to decode complex phenomena and to measure the local actions that can be undertaken: a DT platform has been developed with rGEE code [48].

The Earth Observation can date back to the last two decades thanks to the satellite data (Landsat, MODIS) and the recent Copernicus missions (2017 to now) supporting the land monitoring analysis as in the case of the Basilicata Region with the Medalus system enriching the DT [49].

Under this point of view, in the EO-related disciplines, the availability of mass data accelerates the transition toward fully functioning live Digital Twins. In the case of Appian Park, the capability of the DT to highlight and support analysis of transformation across the seasonal period is reported here. In future, it could be coupled to the HR orthophoto and to the historical data reconstructed to investigate the capability to support semi-automatic data management as in the case of the maintenance yearly actions.

As described in Paragraph 6.5, the GIS DT so far generated and derived from the HR Orthoimages a first rough estimation of the Green areas to be mowed (ordinary maintenance and extraordinary maintenance to free the green areas from brambles that overcame the carriage and sidewalk): the result gave back an estimation of 221,769.5 mq AREA of green vegetation and an AREA of brambles correspondent to ~29,388 mq).

The EVI comparison between these two periods, where June 2022 corresponds to the ordinary maintenance of vegetation without the brambles costs, and October 2022 corresponds to the bramble maintenance extraordinary costs, opens the possibility of linking the two scale levels (10 m Spatial Resolution of the Sentinel-2 and 5 mm of the Orthoimage DT), to monitor vegetation trends (Figure 20).

This could be ideally enriched with the analysis of the Agro-Romano landscape, nowadays facing Climate change threats. The analysis could exploit the use of Cutting-Edge EO Cloud Computing Techniques, such as Google Earth Engine, to extract the phenological trends across the last decades, thus, linking them to historical maps and pictures documenting the past (Figure 21).

### 8.2. Immersive DT Enriched by Informed Models for Visitors’ Well-Beings

Undoubtedly, the generation of 3D high-resolution (HR) textured models thus far represents a significant and notable endeavour, considering the extensive quantity and quality of data involved. However, it should be acknowledged that these models serve as a starting point, or “time zero,” from which further progress and development can be made.

As highlighted in various research studies, such as the VIGIE report, it is widely recognised that achieving a high level of 3D model quality comes with substantial costs. Therefore, it is essential to establish a sustainable environment that can effectively manage the complexity of the data. This involves updating the data and enriching the 3D models for different purposes, from preservation actions to enhancing visitor experiences.

Unlike the ongoing advancements in geoportals, geoservices, and Earth Observation (EO) Digital Twin domains, the cultural heritage domain exhibits a gap in sustainable platforms managed by curators, Protected Area Administrators (PAs), and visitors. This gap pertains to the costs and capacity-building required to sustain such platforms. Recognising that a 3D model without integration into a live data management system risks stagnating the representation of reality and its inherent richness is crucial. Consequently, this limits the analysis, transformation, and enrichment a Digital Twin could facilitate. To address the risk of leaving historical and current data collections and models stored away and inaccessible, ongoing experimentation has been conducted to explore the use of interactive web platforms. These platforms aim to provide progressively sharing and utilising the collected data, allowing for greater accessibility, analysis, and engagement with cultural heritage resources.

The capability of the XR platform [50] to relate the 3D models has the advantage of storing properly different valuable information for understanding the different characteristics of the object of interest, organised in a spatial and semantic database including the construction techniques and materials derived from the HBIM nodes. In the case of the Claudius Anio Novo Aqueduct digitized section, part of the PAAA nodes of the DT, the 3D HBIM models are related to the information through the stratigraphic units [37].

The reliability of the HBIM model is a crucial topic for those involved in its creation, management, and utilization. It revolves around the digital survey, representation, and interpretation of the model, encompassing aspects such as dimensional consistency, materials, construction techniques, information, observations, and associated information.

The concept of “transparency” plays a central role in assessing model reliability and has been discussed in the literature since 2009 through the London Charter [51]. Later, in 2012, the Seville Principles [52] complemented the London Charter, particularly focusing on the archaeological domain. The Charter’s emphasis lies in computer-based visualizations of Cultural Heritage, where “intellectual transparency” involves conveying the sources used in the model through communication.

The Seville Charter, in contrast, emphasizes “scientific transparency”. It allows three-dimensional visualizations to be accessible and usable to other experts who can verify or challenge the obtained results. To achieve this, the introduction of paradata, alongside metadata, is proposed.

Metadata refers to additional data that provides a detailed and comprehensive description of an object or subject. In the context of the HBIM model, it might include information about the model’s author and the indirect sources used during its construction.

On the other hand, paradata represents ancillary data about the data collection process. For built heritage, this encompasses details about the acquisition and survey methods, such as direct measurements, laser scanning, or photogrammetry.

Once these aspects are well-defined and documented, it becomes possible for others, even those not directly involved in the research, to understand the relationships between sources, interpretations, and the three-dimensional model. This transparency promotes clear and open communication of the research results.

These aspects have been considered, for example, in the creation of the digital twin post-disaster built heritage reconstruction process of Notre Dame de Paris [53]. In this case, the digital twin experiment comprises four key components: physical anastylosis, reverse engineering, spatio-temporal tracking of assets, and operational research. These facets are thoroughly described and combined to support a hybrid reconstruction hypothesis. The digital twin allows the simultaneous unfolding of physical and digital processes, storying diverse information as semantically structured data. The outcomes of the study demonstrate that this modelling approach effectively formalizes and validates the reconstruction problem, leading to improved solution performance.

Semantics and vocabulary are crucial to the Digital Twin enrichment, especially in documenting the richness of construction techniques [54]. The paradata about acquisitions are structured according to the PAAA (MIC-Ministero della Cultura) Cultural Heritage data structure, also inheriting the vocabulary and semantics implemented in past experiences (as a Vault Vocabulary for HBIM and GIS across EU) and on 3D Quality Models managing the complexity. A vocabulary has been undertaken with the PAAA contribution in the cases of the Appian Way and of the Claudius Anio Novus Aqueduct upper mentioned.

Further development of the platform could be exploited in the future implementation of the direction of typical DT applications to the microclimate indoor data management of sensors data network also using dedicated platform as Dasher Autodesk platforms [55]. As well as deploying innovative services collecting crowdsourcing images documenting interests, decays, or threats by the visitors themselves.

In particular, thanks to the development of XR platforms such as Twinmotion, Unreal Engine 5, and Unity, it has been possible to create a virtual environment in which digital models can be imported and updated in the cloud, offering a more interactive and updatable version of the environment in real-time according to needs project or update of the current status of the Appian Way.

Thanks to the real-time synchronisation between modelling applications such as McNeel Rhinoceros and Twinmotion, it has been possible to develop an environment which can be implemented and customised at any time. In particular, when VR becomes a DT implementation, it offers several significant benefits for understanding the Roman infrastructure. Firstly, it provided cloud accessibility, allowing users to access and experience the Appian Way directly through a web-VR implementation without needing specific hardware or software installations. This accessibility ensured that a broader audience, including clients, stakeholders, and the general public, could easily explore and comprehend the Roman infrastructure and the project without encountering any barriers. At the same time, the DT provided an immersive experience, enabling users to navigate and interact with the virtual environment. Users could establish a sense of presence within the research case study through VR headsets or a computer or mobile device. This immersive quality enhanced their understanding and engagement with the project, as they could visually and experientially explore its various aspects.

Secondly, DT enhanced heritage understanding, a fundamental aspect of the uniqueness of the research case study. Users could perceive and comprehend the environment’s scale, proportions, and spatial relationships by virtually exploring the DT in three dimensions. This level of immersion allowed for a deeper understanding of the space’s spatial and heritage qualities, flow, and functionality, which could be challenging to grasp through traditional 2D representations.

Furthermore, DT facilitated real-time design evaluation, enabling architects, designers, and clients to assess design decisions as they unfolded. In real-time, users could visualise and experience different design options, material choices, lighting conditions, and spatial configurations, becoming an updatable model able to support designers and users in figuring out possible configurations in different periods.

Moreover, multiple users could simultaneously access and explore the virtual environment, allowing synchronised reviews, discussions, and feedback exchanges. This collaborative aspect facilitated effective communication, shared understanding, and alignment among professional and virtual tourists, leading to better-informed design decisions.

Lastly, virtual tours and interactive presentations of architectural projects could be easily shared online, allowing potential investors or the general public to experience and appreciate the design before its physical realisation.

The ancient view comparisons (middle). The Appian Way Canina ‘concept idea of the Appian Way in the Roman era (right)’, the ruins of the sepulchres centre and left (Luigi Canina). The geometric navigation across the Appian Way portfolio sections contextualises each detail and respects the position and the related information in a virtual immersive gran tour rediscovery of the Appian Way secrets (bottom).

The interactive virtual experience enables the comparison of historical and current views, aiming to disseminate the represented values to the general public and promote an understanding of the intertwined functions that have evolved over the centuries. For example, the Appian Way served military and funerary purposes, with families constructing tombs and mausoleums along its path. The landscape surrounding the road has gradually accumulated layers of history, becoming an essential aspect of its identity for both enjoyment and conservation. Figure 22 and Figure 23 illustrate the outcomes of intertwined multi-temporal, multi-scale, and multi-layered Digital Twins. The implemented web interactive eXtended Reality (XR) platform showcases a sample development in the context of the Appian Way, providing support for a virtual tour that facilitates the understanding of multi-temporal experiences.

Through immersive navigation across the centuries, users can explore and document the historical views of the Appian Way, observing its progressive transformation enriched by various components. The Digital Twin of the Appian Way is addressed to manage the flux of information among the physical and virtual models to better support decision making: it is the case of the design project addressed to foster social well-being within health infrastructure that is using the 3Dmodels for the design purposes and XR experiences [56]. The Digital Twin has the ambition to create a bridge between the past, present, and future using the 3D sections as a grip-scheme where to convey the flux of information by the XR platform.

The XR Digital Twin will significantly contribute to the comprehension of multi-layered and multi-temporal informed models, thus enhancing visitor awareness. Visitors will be able to engage with different layers and components, gaining insights into the materials and construction techniques employed in the Appian Way. This knowledge, which is often unfamiliar to the general public, encompasses features such as the *Pavimentum*, *Basolatum*, and *Crepidines* of the Roman road, as well as the *Macére* walls constructed by L. Canina to demarcate the state-owned area. By preserving the remains on-site, the Appian Way becomes an open-air museum. Each of its 117 sections, spanning 100 m, becomes a source of knowledge, revealing the richness of the Appian Way’s context and providing an immersive model coupled with relevant information. This allows for an informed virtual-physical contemporary “gran tour” experience, offering a deeper understanding and appreciation of the site.

### 8.3. Digital Twins Enriched by Non-Metric Perspective View Comparison

The IV mile of the Appian Way offers a first demo from the Tomb of Marco Servilio Quarto to the high relief with a male figure. Research is ongoing since other pictures, texts, and publications can progressively enrich it. The Appian Way has always fascinated artists, architects and tourists for its history, monuments and landscape. The Appian Way was approximately 4.10 m wide to allow easy circulation in both directions and had about 3.10 m wide sidewalks. Although it was mainly built for military reasons, it became a funerary road since many noble families built their funerary monuments on the roadsides.

However, around the 4th century, the Appian Way began to be affected by the decline of Rome due to wars and invasions, which lasted until the 6th century. At that time, the Church started to acquire properties on the side of the Roman ‘Campagna’ mainly from the emperors. During the Grand Tour, the Appian Way became one of the sites to be visited by artists, poets and scholars interested in Roman history and monuments. However, some people rent the land at the side of the Appian Way to make excavations to find deposits to sell in the antiquities trade market. Since buildings were not protected, many of them were demolished.

In the 19th century, thanks to the work of artists and architects such as Canova, Valadier and Canina, the Appian Way was studied and protected. Canova was the first to understand the importance of keeping the findings on site, not placing the inscriptions or marble pieces in museums. This was achieved by building a scenography with architectural brick walls where deposits found during the excavation were placed to recall the Roman monument. One of the first examples is the Tomb of Marco Servilio Quarto at the fourth mile. By doing this, the Appian Way became an open-air museum. If the work by Canova was mainly focused on some monuments, it was Canina who worked on the fourth mile of the road, making it an archaeological boulevard. He restored the monuments and identified the limits of the archaeological boulevard by building the Macère, small walls on the sides of the road, bringing to light the polygonal blocks of mafic lava of the pavement and the *Crepidines* (the margins of the roadway).

In the third volume of his ‘Memorie Enciclopediche’ (1815) [57] (pp. 135–136), Giuseppe Antonio Guattani (1748–1830), archaeologist, writer and professor, describes the finding of the monument by Canova. He describes the several fragments found by the artist, such as basis, capitals, pillars, columns, and two statues which he thinks are Marco Servilio and his wife. Guattani regrets the difficulty in reconstructing the monument due to the state of conservation of the deposit. He also describes the different hypotheses regarding the shape and elevation of the tomb but does not describe the work by Canova, who will place the fragments on a brick wall with a brick and tuff stones base. The deposits (capitals, inscriptions, marble decorations) are placed in a way that gives us an idea of the monument, still open to interpretation.

Plate XVIII by Canina is referred to this section of the road. On the plan and under the scene, the tomb is called ‘Servilii’ to mean the surname of Marco Servilio and his Gens Servilii. Strangely enough, in his book, Canina does not mention the fragments of statues depicted by Guattani and does not even draw marble decorations or other findings as he has done for other tombs [58] (p. 15).

Canina represents the monument along the Appian Way without giving it much space inside the book, probably because it was not one of the resounding discoveries. Suppose the identification of the tomb is right on the scene of the road reconstruction. In that case, the tomb is represented as a funeral chapel with a rectangular plan raised on a podium. The temple has a portico with four Corinthian-order columns and an architrave. The roof is a step pyramid with a statue on top. The views can be compared to the 3D textured twin model allowing operators, curators and visitors to understand the ancient use, the state of conservation and the transformations that occurred as in the landscape case (Figure 24). After the great work of Canina for the documentation of the state of preservation of the monument is worth mentioning the role of photographs. The photographs by Pompeo Bondini (1828–1893) were shot in the same year of the completion of the Canina’s work, in 1853. Bondini is one of the first photographers in Italy and published ‘Della Via Appia e dei Sepolcri Romani’ that contained 42 salted paper prints from calotype negatives—one of the first publications in which photographs are an integral part of the written text [59]. The picture takes the monument from its right side. Two things leap to the eyes: other fragments based on the monuments and no vegetation behind the *Macére*. These fragments were depicted even by Canina, but they are missing from a print of the 19th century by F. Rinaldi [60] (p. 96).

The vegetation along the Appian Way has changed since the 19th century. The secular pines, praised by Ottorino Respighi and the Cypresses, were planted by Antonio Munoz, Regia Soprintendenza ai Monumenti’s inspector between 1909 and 1913. Three photos preserved in the Gabinetto Fotografico Nazionale and available online, thanks to the archive digitisation by the ICCD (Istituto Centrale per il Catalogo e la Documentazione) [61], depicted the monument. The Gabinetto Fotografico Nazionale was established in 1895 to document Italy’s cultural and landscape heritage, which was united 34 years earlier. It understood the importance of photography right after a few years of its invention (1839) as a perfect tool for mapping cultural monuments and sites. The three photos date back to 1951 and show a different situation for the monument: pines have been growing, and vegetation is over the monument since plants are on its top. The pictures are taken from a quiet distance, so it is impossible to see the state of conservation of the materials, which seems better than today (Figure 25).

## 9. Results and Discussion

In general, the survey of infrastructure is carried out on a map scale 1:2000/1:1000, with the archaeological survey at no less than a scale of 1:50. The bet of this multi-sensor survey was to reason with the approach of the archaeological survey in an extension typical of an infrastructure. To document and read the whole 12 km stretch metrically with the precision that is asked to the survey for the documentation of archaeological artefacts. All are made more difficult by the complexity of an infrastructure where details change from meter to meter, from centimetre to centimetre. You see the discontinuities at times readable, at times just mentioned, and at times almost completely disappeared between the break line of the roadway and *Crepidines*. This material consistency variability does not allow thinking about the classical discretised representation for transversal and longitudinal sections, their fronts, and orthogonal projections, as in the case of the planimetric orthomosaic projections. To document the complexity of the interventions that have occurred over time and the current conservative state, it became necessary to move to a three-dimensional control of the whole model so that the different interventions by the PAAA could be planned.

Managing a texturised High-Resolution model goes beyond the simplified CAD management, which does not handle texturised 3D models. It allows a full-round metric and semantic control along the entire surveyed section. This result was made necessary by the integration of multi-sensor surveying techniques.

The optimisation of the geometrical grip-schemas, both the 360° videos from the spherical cameras and the MMS point clouds, has allowed acquiring a massive result by exploiting the flexibility of survey in areas of difficult coverage, limiting drift effects with closed loop patterns for both sensors. The measurement of GCPs with RTK-GNSS positioning was fundamental for the georeferencing of TLS and MMS data and photogrammetric data acquired from three sensors (drone, 360° and Nikon camera); framing the data acquired by the different sensors (MMS, TL, 360° and frame cameras) in the national cartography and reduce drift effects and deformations of MMS sensors and photogrammetric blocks. In the future, further tests will be carried out concerning the influence of the number and distribution of GCPSs on georeferencing accuracy, optimal planning of the MMS and 360° image trajectory to reduce drift effects and optimise scanned areas, a fusion of the data acquired by different sensors. The experimentation of acquisition and processing aimed at documenting the vegetational layer is very topical and opens up new areas of investigation, especially if addressed to document the degree of conservation loss or decline of biodiversity, especially when integrated with Earth Observation techniques.

A further discussion element concerns the transition from 3D Quality Models managing the complexity [62] into the Digital Twin (DT) interpretations. The DT of the Appian way has been undertaken under different points of view upper described. Notably, the experimental XR web interactive platform is conceived to support a live Digital Twin for curators, operators and visitors bridging reality and virtual objects, with the aim of reducing ICT costs of DT platforms. It is ready to support future developments, including fluxes of data from citizens as well as IoT sensors.

## 10. Conclusions

Achieving the accuracy of a global texturised 3D model at the scale of an infrastructure-archaeology requires increased processing time. Data optimisation according to the objects and layers of the Appian Way (infrastructure with the archaeological detail of the pavement, as the parts in *Basolato*, the documentation of the *Crepidines* and their stones, or the *Macére* border walls, of the archaeological artefacts, such as the tombs, up to the vegetational layer): this refers not only to the acquisition phase but also to the post-processing phase, such as differentiated cleaning of point clouds, both in photogrammetric processing and in the data merging and management of TLS clouds with MMS clouds. The results obtained allow usability by experts, designers and the PAAA to support a timely and targeted planning of ordinary and extraordinary maintenance works (see the mapping of mowing areas in the roadway or the aggressive vegetation that gradually leads to the disappearance and loss of consistency of *Crepidines*); both to support the design of the slow mobility plan, of service areas to support better usability along the entire stretch and not only concentrated on the few current rest areas.

Managing a global 3D model also requires changing habits to use 3D models obtained as a daily rule, not as a one-time scenic view. From this point of view, both the client, the PAAA, and the designers have understood the importance of starting to make a watershed towards habits that will facilitate the impact assessment and quantification of the works and see the use of HBIM as an additional three-dimensional computing and management element. At the same time, their cost is amortised, thanks to their reusability for communication purposes through extended Reality tools. Reasoning in a multi-temporal Digital Twin extended to various historical phases allows you to capture in space 4D information and archival documents, to easily access such complex informative models; at the same time, it sensitises visitors to informed processes aimed at the modern fruition of the Gran Tour revisited in a hybrid key in new experiences made by physical and virtual immersive paths where each detected section can become the learning base of narrative storytelling and in-finite research.

## Figures and Tables

**Figure 1 sensors-23-08556-f001:**
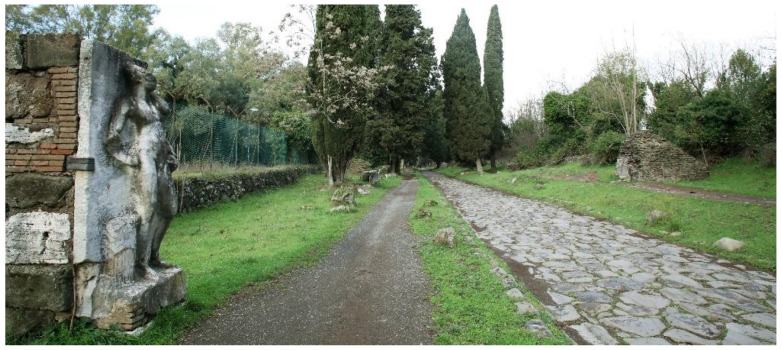
The Appian Way at the fourth mile 10.7–10.8 km: Statue of a heroic figure.

**Figure 2 sensors-23-08556-f002:**
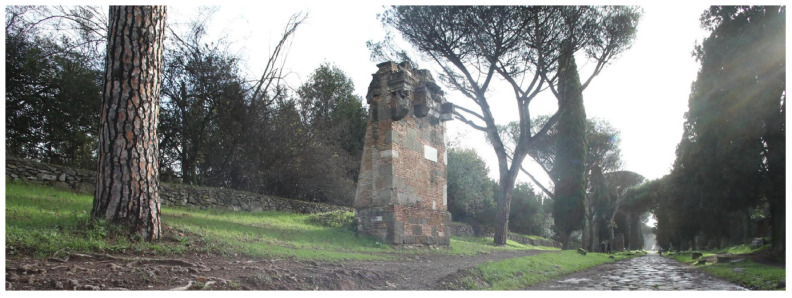
The Appian Way at the fourth mile 10.7–10.8 km: the Tomb of Marco Servilio Quarto.

**Figure 3 sensors-23-08556-f003:**
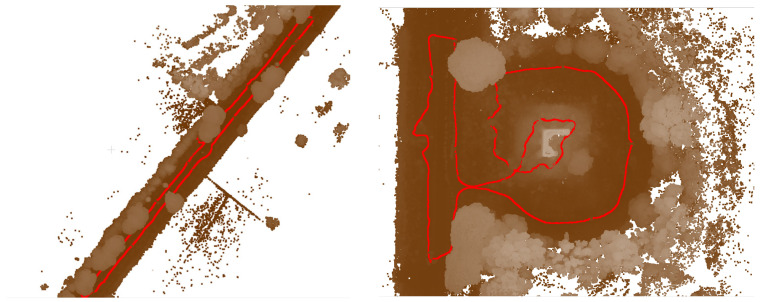
Example of point cloud acquired with the MMS system: the point cloud colorised according to the height; the trajectory in red.

**Figure 4 sensors-23-08556-f004:**
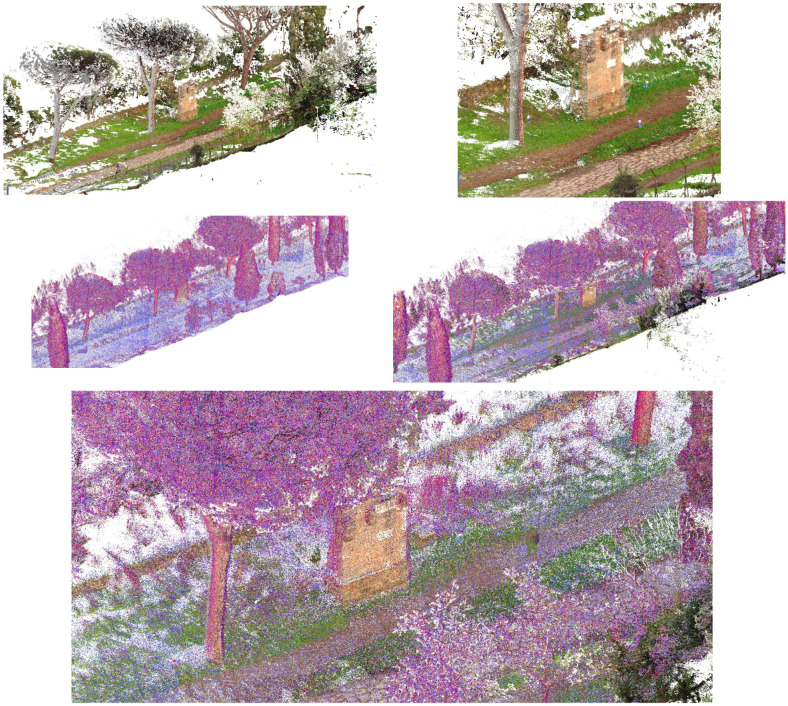
Section 10.7–10.8 km with the Sepulchre of Marco Servilio. The TLS Faro Focus3D and a detail of the spheres and GNSS targets used for the data registry (**top**). The MMS cloud points (**centre left**); the merged TLS FARO FOCUS and MMS cloud points (**centre right**); a detail of the merged clouds (**Bottom**).

**Figure 5 sensors-23-08556-f005:**
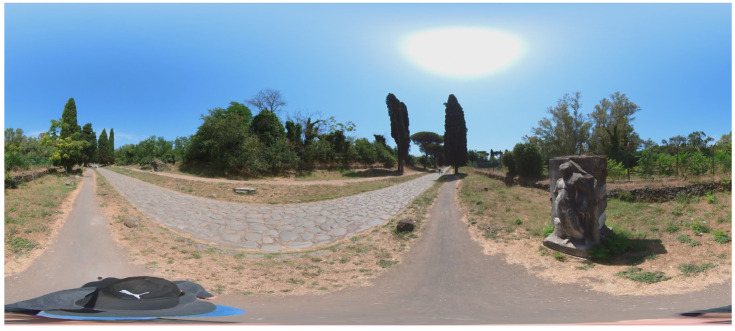
Example of 360° image acquired for the fourth-mile test area.

**Figure 6 sensors-23-08556-f006:**
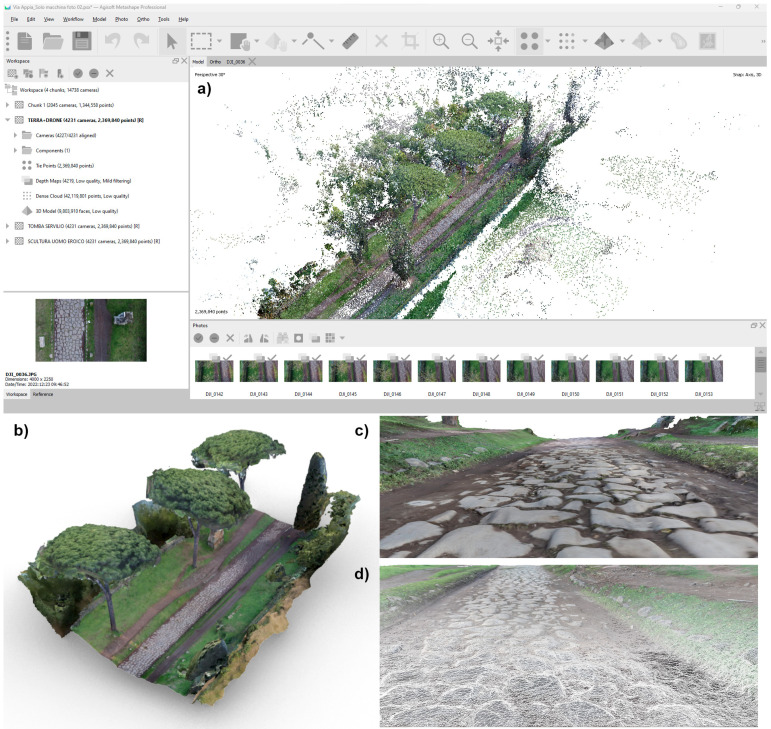
The multi-sensors photogrammetric process. (**a**) Photogrammetric image block (UAV and cameras) optimised for the Digital Twin of the focus area of the Sepulchre of Marco Quinti Servilio; (**b**) the 3D textured mesh model; (**c**,**d**) detail of the 3D textured mesh model of the *Basolatum* paved surface.

**Figure 7 sensors-23-08556-f007:**
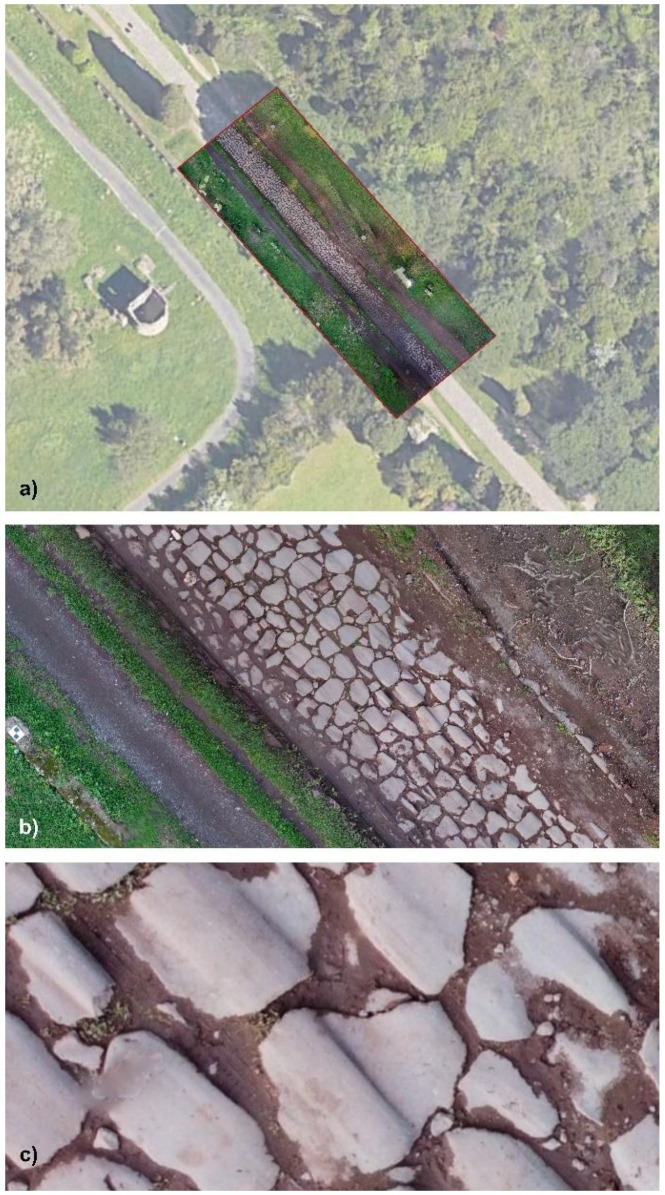
The Orthomosaic output obtained at different scales and resolutions (**a**–**c**) and the horizontal plane to be managed within CAD tools.

**Figure 8 sensors-23-08556-f008:**
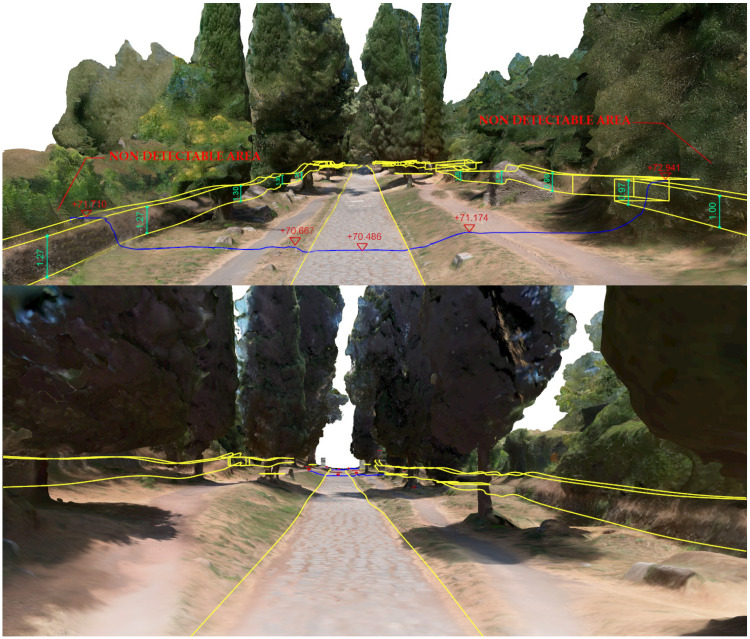
3D immersive high-resolution textured models documenting the multi-stratified Appian Way layers. The immersive 3D model representation of the multi-stratified Appian Way layers: the infrastructure, with *Basolato Pavimentum* and *Crepidines*, the *Macére* drywall on the state’s own borders, the landscape, vegetation, the architectures, archaeological artefacts, and remains. The terrestrial laser scans derived the transversal profiles.

**Figure 9 sensors-23-08556-f009:**
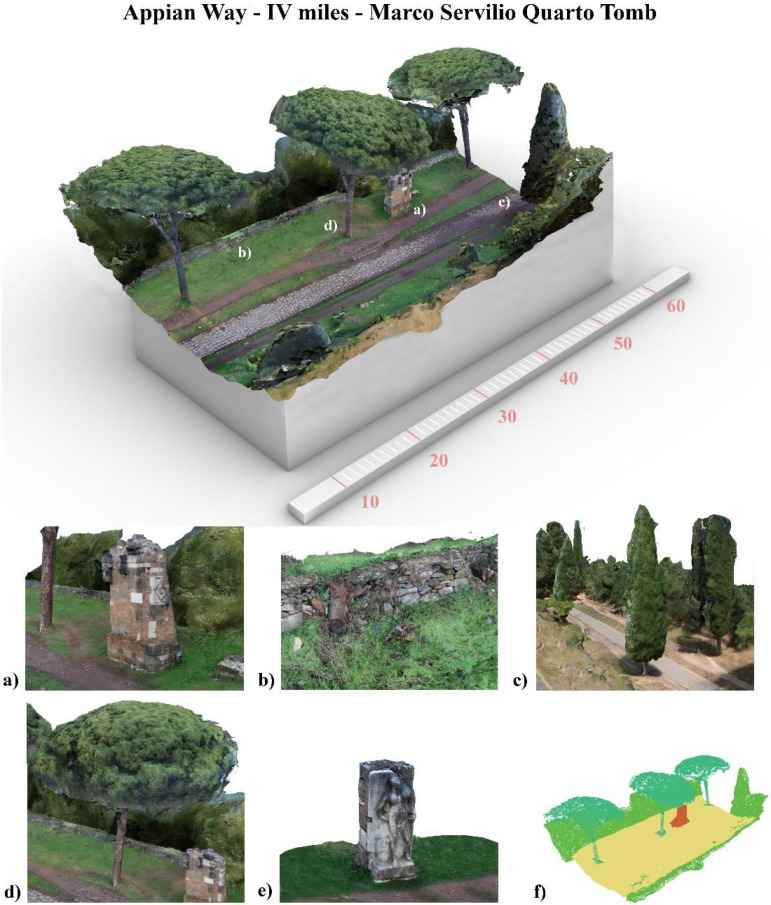
3D textured grip models of the Appian Way infrastructure section (the photogrammetric output management into pure modellers (as Rhinoceros McNeel©). Top: The textured mesh model section of about 60 m of the Marco Servilio Sepulcher: (**a**) Marco Servilio Tomb; (**b**) *Macére;* (**c**) *Cypress*; (**d**) *Pinus Pinea*; (**e**) statue of heroic figure; (**f**) the mesh model is the result of different photogrammetric blocks—Appian Way carriage (UAV survey, yellow), Marco Servilio Tomb (ground camera, red), *Pinus pinea* (UAV survey, light blue), vegetations (360° cameras, green).

**Figure 10 sensors-23-08556-f010:**
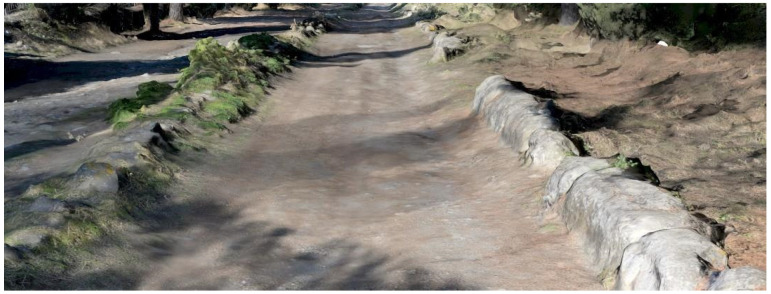
One of the first miles, where the *Basolatum* paved surface is not present, while *Crepidines* are visible.

**Figure 11 sensors-23-08556-f011:**
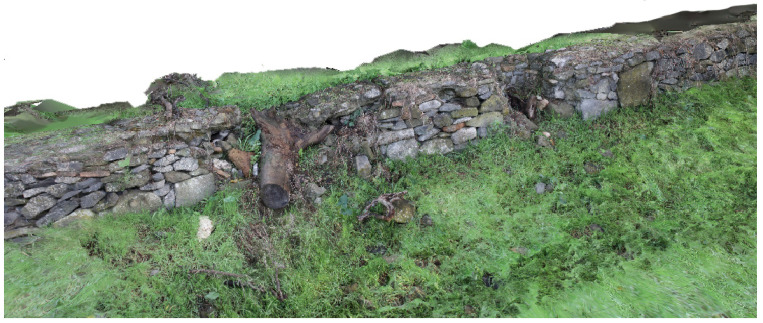
The *Macére*: a detail of the 3D textured model documenting a portion of the masonry wall damaged with clashed portions and vegetation.

**Figure 12 sensors-23-08556-f012:**
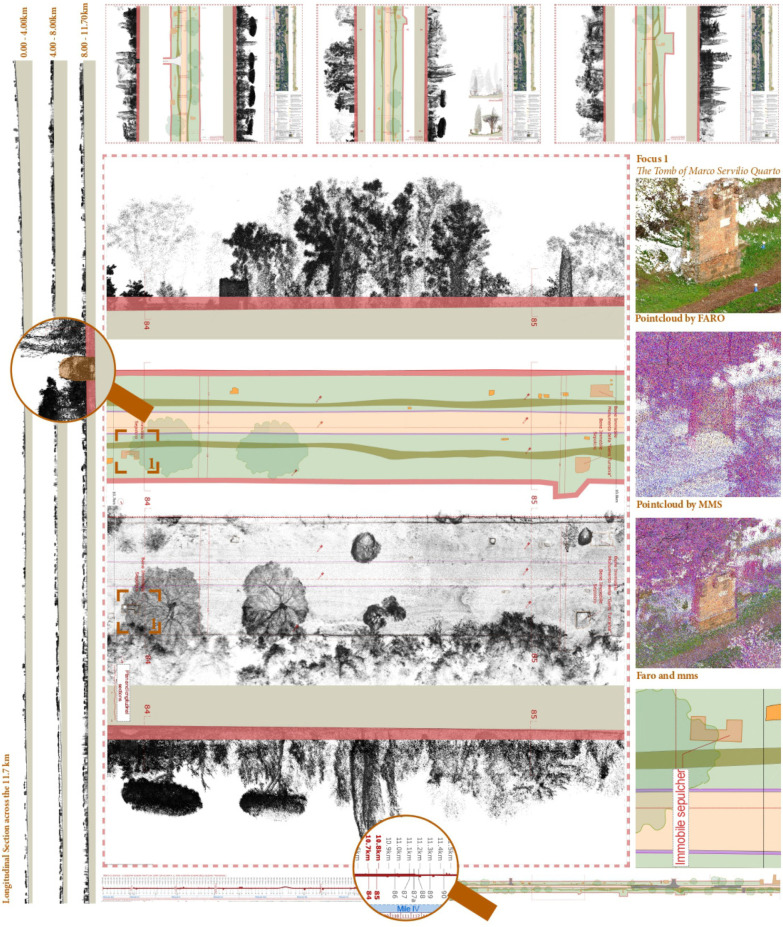
The Appian Way CAD Portfolio (the sequence of 4 adjacent Panels 107 108 109), (**upper**). (**left**) the vertical Longitudinal Section (0–4 km, 4–8 km, 8–11.7 km), (**left**). The grid schema of the Portfolio Panels, with the number of the Panels, the geographic coordinates, the Miliarum, and the vertical Transversal Sections. A detail of Section 10.7–10.8 km (Panel Tav. 108 with the geometric restitution, the Panel 108b with the cloud points vegetation layers for the design purposes), (**centre**), and details (**left**).

**Figure 13 sensors-23-08556-f013:**
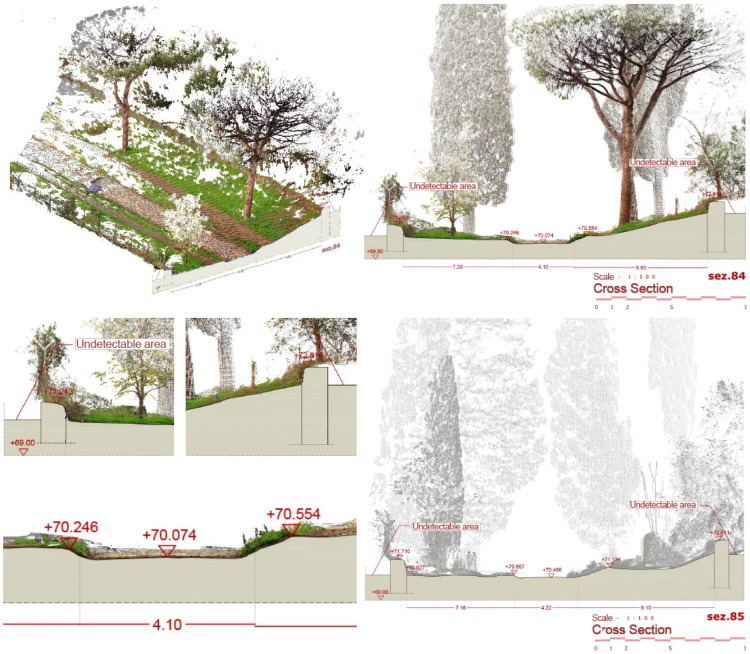
The Appian Way CAD Portfolio. Detail of the Transversal vertical sections 84 and 85 derived from the cloud points models (TLS, MMS and Photogrammetric models).

**Figure 14 sensors-23-08556-f014:**
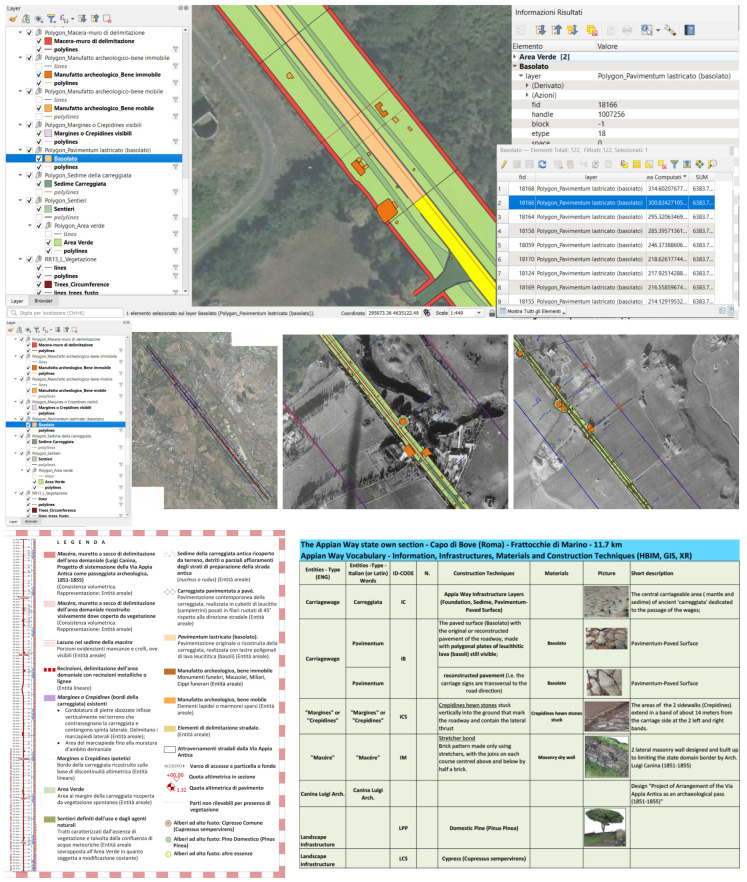
The Appian Way GIS Geopackages data management legends and vocabulary (the GIS legends are in italian being official documentation for the PAAA, the vocabulary represents the english translation). Detail of the APPIAN GIS State-own Section 108 (10.7–10.8 km) with the Marco Servilio Sepulcher (Archeological unmovable artefact layer) and the Vertical Transversal Section n. 85 (**upper left**): the Layer of *Pavimentum* (*Basolato*) with the Attribute Table Area computation for maintenance purposes and the SUM of all the areas 6380 m^2^ (**center** and **2nd left**). The superimposition with the historical maps documentation (**middle right**): the GIS GEOPKG layer entity management, cataloguing all the Appian Way features; the Orthoimage of Lazio Region© courtesy of Regione Lazio details (**1st left**) with in Blue, the longitudinal and transversal vertical sections (**middle left**); the georeferentiation of the 1958 orthophoto (**2nd left**). The CAD legend and the external Vocabulary shared among GIS and HBIM systems (and XR) (**bottom**).

**Figure 15 sensors-23-08556-f015:**
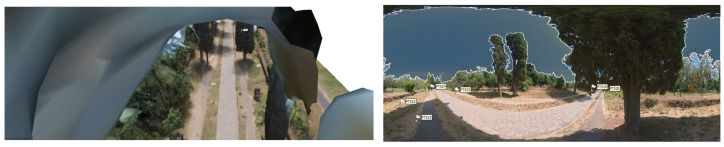
The model of the Appian Way was generated without using pre-processed Dense Cloud data from both the 360° and Drone surveys. In the visual representation on the left, one can observe the disturbances caused by the presence of the sky in each frame. These disturbances are clearly visible and impact the overall quality of the model. On the right, masks have been applied to each 360° image to address this issue.

**Figure 16 sensors-23-08556-f016:**
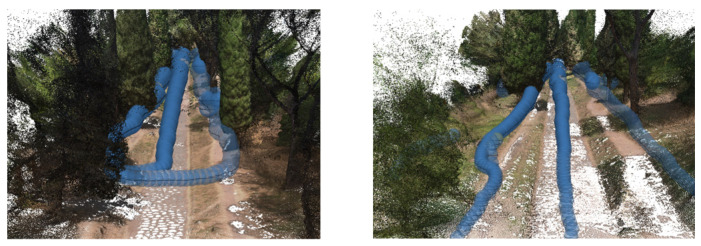
After applying filters and masks during the post-processing stage, the dense cloud derived from the 360° data exhibits notable improvements (the methodology of field acquisition is also evident). These enhancements result from the post-processing techniques employed, which include filtering and masking procedures.

**Figure 17 sensors-23-08556-f017:**
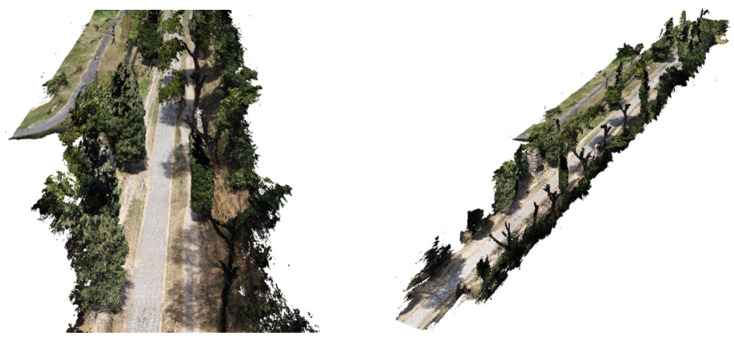
The pre-processed Drone Dense Cloud, with disturbances removed and sky coverage, masked out, yielded a Raw Dense Cloud with 33,180,126 points. This detailed and accurate representation captures the essence of the surveyed area with precision.

**Figure 18 sensors-23-08556-f018:**
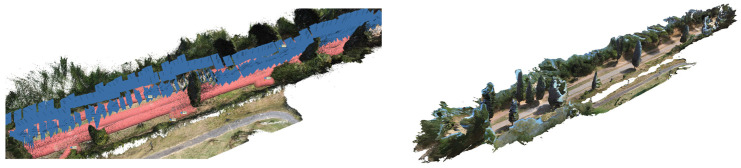
(**left**) the final Dense Cloud product is achieved by aligning and merging the post-processed Dense Cloud of the 360° images (in pink, the 360° frames) and the cleaned Drone Dense Cloud (in blue, the drone frames). (**right**): the two chunks are aligned with Point Cloud assets settings and then merged to create a unified representation of the surveyed area.

**Figure 19 sensors-23-08556-f019:**
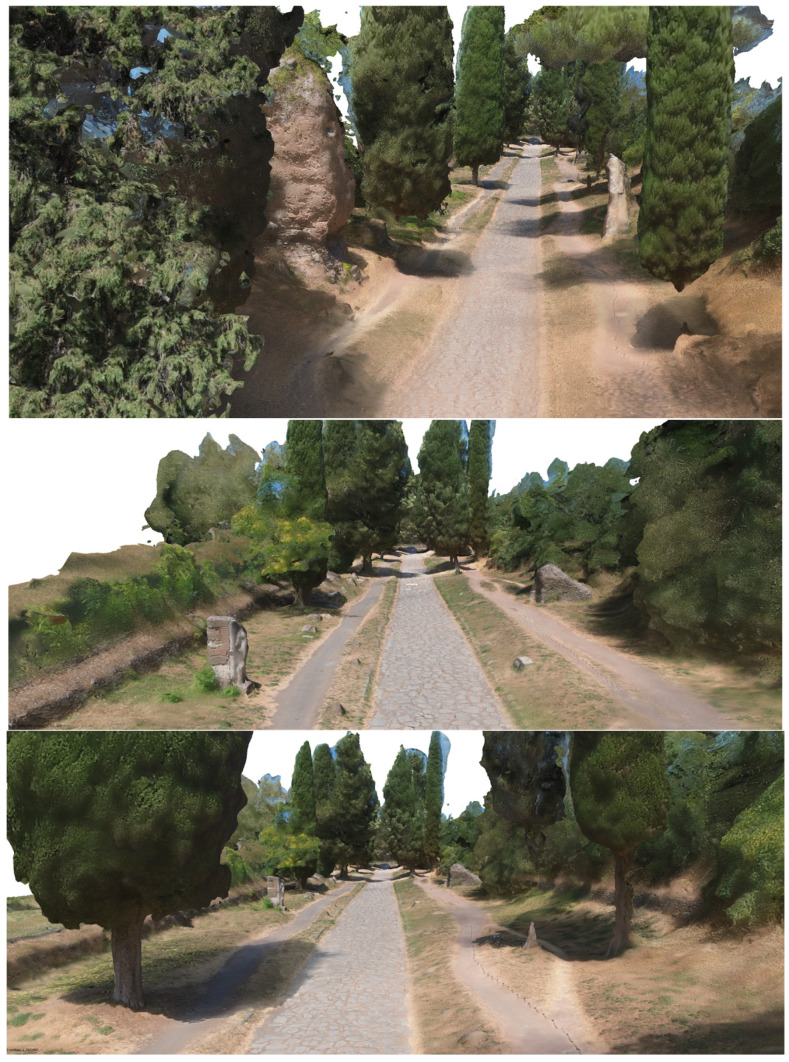
The immersive 3D landscape documentation of the Appian Way 3D reconstruction: the vegetation component.

**Figure 20 sensors-23-08556-f020:**
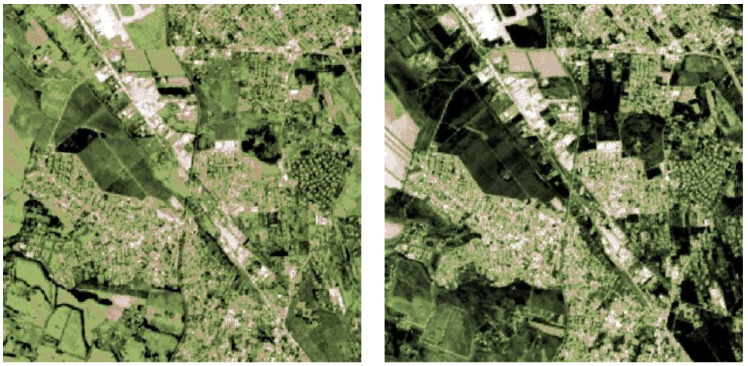
DT sequence of the Enhanced Vegetation Index (EVI) comparison for the Area of Interest (AOI). Projected resolution: 7 m/px, UTM 33N (EPSG:32633). On the left, June 2022, Copernicus Sentinel-2 L2A, 19 June 2022 10:09:12 UTC, 0.0% Cloud Cover, 33TUG; on the right, OCTOBER 2022, Copernicus Sentinel-2 L2A, 17 October 2022, 10:09:07 UTC, 6.8% Cloud Cover, 33TUG.

**Figure 21 sensors-23-08556-f021:**
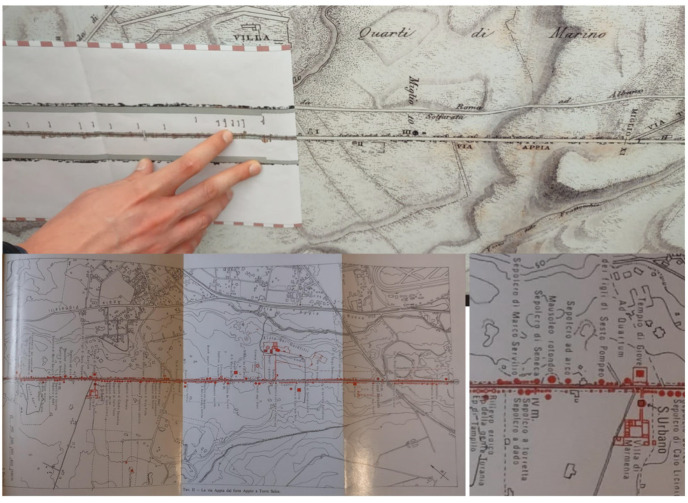
The Appian Way analogic geographic description with the monument’s archaeological artefacts and sepulchres positions. (**Top Right**) the Historical Map of Canina 1853 1:5.000 (a printed version viewable by the public on the Appian Way at the Civic Number Capo di Bove N. 122, © Photo credit PAAA. (**Top Left**) the analogic Twins comparison: the current GIS maps survey superimposed to the Canina Map (ABC-GICARUS and PAAA); (**Bottom**) L. Quilici Tav II—La Via Appia dal forte Appio a Torre Selce (1977) and the detail of the Sepulchre of Marco Servilio, © courtesy of L. Quilici.

**Figure 22 sensors-23-08556-f022:**
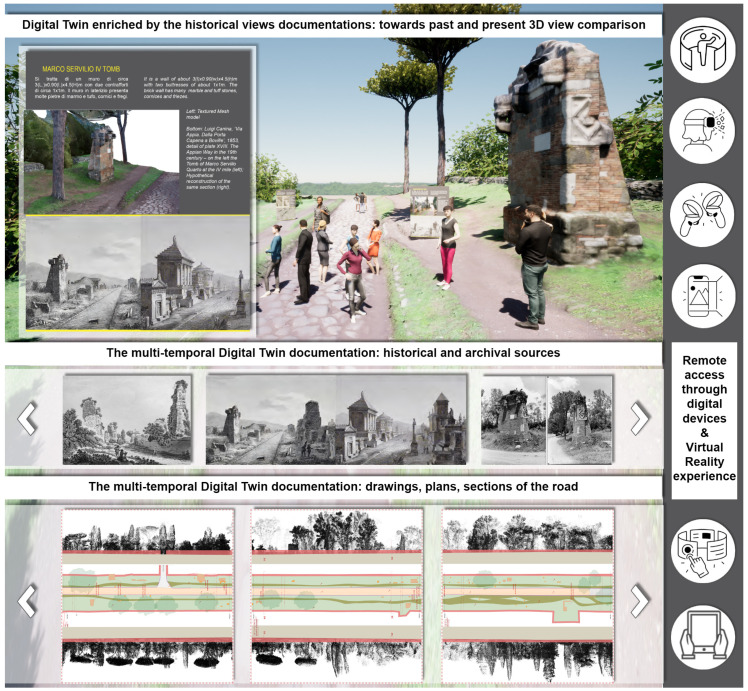
The web interactive eXtended Reality platform implemented and a sample of development in the Appian Way: it supports a virtual tour understanding the multi-temporal fruition, with the immersive navigation across the centuries documenting the historical views of the Appian Way, the transformation progressively enriched by the vegetation and landscape components, the Marco Servilio Tomb explanation. The icons on the right side refer to the different devices and environments of XR remote access.

**Figure 23 sensors-23-08556-f023:**
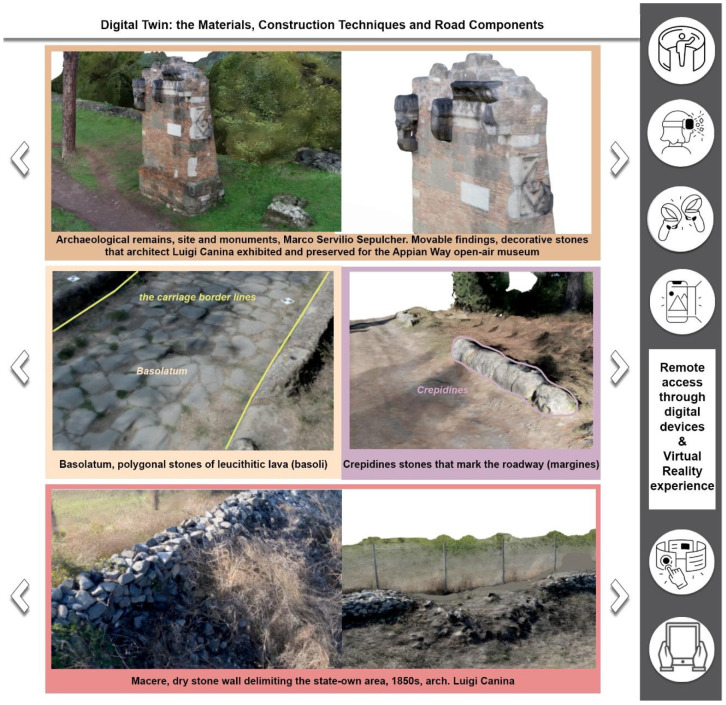
eXtended Reality multi-layers multi-temporal multi-scale informed models rising awareness among visitors. Visitors can read the different layers and components, understanding the material and construction techniques of the Appian Way, which is generally unknown as the *Pavimentum*, the *Basolatum*, the *Crepidines*, and the *Macére* walls.

**Figure 24 sensors-23-08556-f024:**
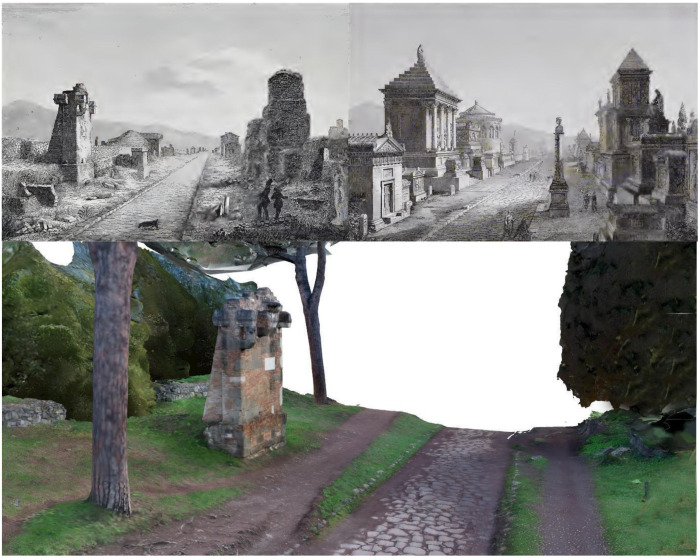
(**Top**) Luigi Canina, ‘Via Appia. Dalla Porta Capena a Boville’, 1853, detail of plate XVIII [58]: the Appian Way in the 19th century—the Tomb of Marco Servilio Quarto at the IV mile, as he found it (**left**); and the hypothetical reconstruction of the same section, made by Canina, a sort of conceptual map he called the ‘Idea of the Appian Way’, imagining the state of the Appian Way at the Roman Era (**right**). (**Bottom**) Multi-temporal Digital Twin documentation: the Canina open-air archaeological museum layer and the 3D view textured model comparison.

**Figure 25 sensors-23-08556-f025:**
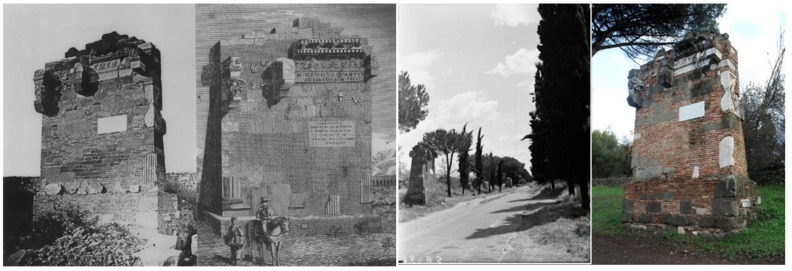
An example of the collection of Photos Archives in the DT. Photo by Pompeo Bondini, 1853 (**1st left**) [60] (plate VIII)—fragments are placed on the basis; Print by F. Rinaldi, 19th century (**2nd left**) [61] (p. 96); The Tomb of Marco Servilio Quarto, Gabinetto Fotografico Nazionale, 1951 (ICCD Inventory N. E058492) [62]; notice that trees, *Pinus Pinea and Cpuressos Sempervirens*, are smaller than today (**3rd right**); the 3D current state of conservation and vegetation attack (**4th right**).

**Table 1 sensors-23-08556-t001:** Comparison of the different surveying strategies in terms of input data, the accuracy of the final output, GBs, and effort of the acquisition stage.

Sensor	Section	GygaByte	No. of Acquisition	Accuracy	Output	Acquisition Days/Hours/Operators
GNSS-RTK	0–12 km	-	567	±2.0 cm plan ±3.0 cm height	GCPs and CPs	-
Ground frame camera (Nikon D610)	0–4 km + focus 11.7 km	~120 GB	no. ~6000 pictures	±2.2 cm on GCPs C.P.s ± 2.5 cm	Orthophoto 1 mm reduced to 5 mm	5 days/1 operator
Ground 360° camera (Insta360 ONE X)	0–4 km + focus 10.7–8 km	~225 GB (~2.5 GB each video)	no. ~15,000 frames extracted (5.7 k)	±2–7 cm on GPSs	3 D textured mesh model	5 days/1 operator
UAV test (DJI Mini 2)	10.7–8 km	~2.62 GB	460 pictures	±2–7 cm on GPSs	Orthophoto 1.0 cm (height 12 m) 1 mm height < 1 m	1 h/1 operator
MMS (GeoSLAM Zeb Horizon)	0–12 km	~53 GB	no. 55 scans	±5–10 cm on GCPs	Point cloud	9 days/1 operator
TLS (Faro Focus 3D)	0–12 km	~170 GB	264 scans	±1–3 cm on GCPs registry ± 3–5 mm locally	Point cloud	9 days/2 operators

**Table 2 sensors-23-08556-t002:** Optimisation of the photogrammetric image blocks in the function of the different Appian Way layers components.

Photogrammetric Image Optimisation	Infrastructure Layer	Infrastructure Archaeology	The Architectural Layer (Unmovable Objects)	Landscape Layer
360° image blocks (+Drones tests 5–10 m)				Landscape documentation (*Pinus pinea* and *Cypressus*)
HR Ground image data (+ UAV ground 1–5 m where available)	Carriage wage geometry + Sidewalks geometry + *Macére*	*Pavimentum* (i.e., *Basolatum* documentation of the stones) *Crepidines* (Stones documentation) Movable Archaeological Remains (stones and decoration)	Monuments and Sepulchres	

## Data Availability

Not applicable.

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
