# Peer review of "Multi-Sensor HR Mass Data Models toward Multi-Temporal-Layered Digital Twins: Maintenance, Design and XR Informed Tour of the Multi-Stratified Appian Way (PAAA)"

_sensors, 2023, doi:10.3390/s23208556_

Round 1
Reviewer 1 Report
The research topic is of interest both for the methodological aspects of the reverse engineering of geometries and especially for the historical and archaeological significance of the initiative. The authors presented valuable research in which they combined different techniques to obtain the reconstruction of one of the most important archaeological sites of the city of Rome in Italy.
The significance of the work is not so much in the reconstruction technique itself but in having been able to combine several techniques to manage such a large and varied reconstructed area with architectural aspects of multiple natures, shapes and sizes. Therefore, I believe that the work is worthy of publication also because many of the adopted methodologies can be a guide for the digitization of other archaeological sites.
On the other hand, my opinion is that three aspects should be improved before a final decision of acceptance.
The first aspect to focus on is the confusion that can be generated by reading the title.
In fact, the article explores the details and the aspects related to the digitization of the Appian Way with particular reference to the detailed reconstruction of textures and three-dimensional geometries. In the title and also in the introduction, reference is made to the concept of a Digital Twin that is much broader than a three-dimensional model, although accompanied by a lot of information that can be explored and interrogated. According to the globally recognized definition of the Digital Twin, in comparison to an accurate 3D model, a Digital Twin has the main features of continuous communication with the physical twin and following up with the physical twin throughout its life cycle (augmenting and completing coming from it). So, strictly speaking, a three-dimensional model with added and augmented information cannot be considered a digital Twin but can be the basis for the construction of a digital Twin.
So, I think it is appropriate that the authors distinguish well a 3D model, even if very rich in information, from a digital Twin and then eliminate the term “Digital Twin” from the title, the keywords and other parts of the paper where it is misused. They can discuss the potential of the use of the model they have reconstructed for the development of Digital Twin that, however, goes beyond the specific purpose of the article. On the other hand, if the authors have actually used the reconstructed models to create proper digital Twins, I suggest that they indicate the details with particular reference to the aspects of physical-digital communication and the supporting aspects of the digital model to the physical one.
The second aspect is related to the idea of the 4D multi-temporal model.
The actual implementation of the 4D model needs to be clarified. How the ancient information is harmonized with the acquired 3D model. How are the photographic documents integrated into the 3D geometries? Are the 3D models updated/morphed/altered in order to include past historical features? This can be a very innovative aspect of the paper, and therefore more details are needed.
The third aspect is related to the idea of the immersive informed model.
This section of the article is vague and could be more specific, and the details could be more accurate. It seems rather a section illustrating the possible applications and uses of the reconstructed 3D models rather than something actually completed in the current research. It is necessary to be more specific and include only what is actually part of this study (tested and prototyped), maybe even with images of implementations and details of the programming and visualization tools used. If the immersive informed model is only an ongoing idea, this section can be moved to conclusions as a possible future development.
Author Response
Dear reviewer,
We sincerely thank you for your invaluable guidance and efforts in reviewing our work. Your insightful feedback has contributed significantly to enhancing the quality and clarity of our research. As a result of diligently following your recommendations, we have made the following improvements to our work – we highlight our replies to the comments in blue (see the attached word file). The major revision have been added to paragraph 8.
The research topic is of interest both for the methodological aspects of the reverse engineering of geometries and especially for the historical and archaeological significance of the initiative. The authors presented valuable research in which they combined different techniques to obtain the reconstruction of one of the most important archaeological sites of the city of Rome in Italy.
The significance of the work is not so much in the reconstruction technique itself but in having been able to combine several techniques to manage such a large and varied reconstructed area with architectural aspects of multiple natures, shapes and sizes. Therefore, I believe that the work is worthy of publication also because many of the adopted methodologies can be a guide for the digitization of other archaeological sites.
On the other hand, my opinion is that three aspects should be improved before a final decision of acceptance.
The first aspect to focus on is the confusion that can be generated by reading the title.
In fact, the article explores the details and the aspects related to the digitization of the Appian Way with particular reference to the detailed reconstruction of textures and three-dimensional geometries. In the title and also in the introduction, reference is made to the concept of a Digital Twin that is much broader than a three-dimensional model, although accompanied by a lot of information that can be explored and interrogated. According to the globally recognized definition of the Digital Twin, in comparison to an accurate 3D model, a Digital Twin has the main features of continuous communication with the physical twin and following up with the physical twin throughout its life cycle (augmenting and completing coming from it). So, strictly speaking, a three-dimensional model with added and augmented information cannot be considered a digital Twin but can be the basis for the construction of a digital Twin.
So, I think it is appropriate that the authors distinguish well a 3D model, even if very rich in information, from a digital Twin and then eliminate the term “Digital Twin” from the title, the keywords and other parts of the paper where it is misused. They can discuss the potential of the use of the model they have reconstructed for the development of Digital Twin that, however, goes beyond the specific purpose of the article. On the other hand, if the authors have actually used the reconstructed models to create proper digital Twins, I suggest that they indicate the details with particular reference to the aspects of physical-digital communication and the supporting aspects of the digital model to the physical one.
Given the length of the paper we didn’t underline the specificity of a DT respect to 3D models. Nevertheless we agree it is very useful to add a brief discussion on the DT interpretation of the terms used in various scientific communities and how we are interpreting the PAAA DT. Chapter 8 has been completely revised and tuned to the DT interpretation. A brief introduction on this discussion has been therefore added [see pp. 29-30-31]
The second aspect is related to the idea of the 4D multi-temporal model.
The actual implementation of the 4D model needs to be clarified. How the ancient information is harmonized with the acquired 3D model. How are the photographic documents integrated into the 3D geometries? Are the 3D models updated/morphed/altered in order to include past historical features? This can be a very innovative aspect of the paper, and therefore more details are needed.
Chapter 8 has been completely revised to emphasize the structures and features of the PAAA Digital Twin. Three topics (and sub paragraphs) have been introduced at three levels:
The Appian Park DT enriched by EO Digital Twins integration and by metric historical data
The study explored the integration of Earth Observation (EO) data into digital twins, particularly in the case of the Appian Park DT. This integration enriched the digital twin with historical metric data, enhancing its ability to capture and represent changes over time. The DT has been undertaken as illustrated below.
see 8.1 EO Digital Twins and the Appian Park DT enriched by metric historical data [pp.31-32]: the sub-paragraph has been added.
Immersive DT enriched by informed models rising awareness and visitors' well-being
The study aimed to create immersive models that prioritise visitors' awareness rising, fostering their well-being in the fruition of physical and virtual spaces thanks to a DT considering emerging paradigms such as interactivity, immersion, and interoperability in virtual reality (VR) and web-VR environments. These models aimed to provide visitors with an engaging and informative experience while ensuring their comfort and satisfaction. Starting from the Villa Quintili web interactive immersive XR DT implemented on the V° Mile [45] an experimental sample is here developed and presented.
see 8.2 Immersive DT enriched by informed models for visitors' well-being [pp.33-36]. the previous paragraph has been turned and better harmonised to the DT focus.
Digital Twin enriched by non-metric perspective view comparison rising comprehension
The study explored the enrichment of digital twins by comparing non-metric perspective views. This approach allowed users to access a virtual tour across the centuries, providing a unique and comprehensive understanding of the object or site represented by the digital twin. In the future crowdsourcing fluxes of images (i.e. on the state of conservation) acquired by visitors could be added to the DT.
see 8.3 Digital twins enriched by non-metric perspective view comparison [pp.37-39] the previous paragraph has been turned and better harmonised to the DT focus.
Furthermore, the XR Digital Twin aims to facilitate the exchange of information between physical and virtual models, bolstering decision-making and design processes that promote social well-being within health infrastructure. This ambitious endeavour seeks to establish a meaningful connection between the past, present, and future by employing 3D sections as a framework to channel the flow of information through the XR platform.
The third aspect is related to the idea of the immersive informed model.
This section of the article is vague and could be more specific, and the details could be more accurate. It seems rather a section illustrating the possible applications and uses of the reconstructed 3D models rather than something actually completed in the current research. It is necessary to be more specific and include only what is actually part of this study (tested and prototyped), maybe even with images of implementations and details of the programming and visualization tools used. If the immersive informed model is only an ongoing idea, this section can be moved to conclusions as a possible future development.
The objective of the XR Digital Twin is to facilitate the exchange of information between the physical and virtual models, thereby enhancing decision-making and design processes aimed at promoting social well-being within health infrastructure. This ambitious endeavor seeks to establish a connection between the past, present, and future by employing 3D sections as a framework to channel the flow of information through the XR platform.
see 8.2 Immersive DT enriched by informed models for visitors' well-being (see the previous note, as upper described)

Reviewer 2 Report
This is an excellent paper and deserves publication as is.
I have only a very very minor suggestion. On line 177, authors mention “the National Recovery and Resilience Plan” and in line 178 they cite the same with the Italian acronym PNRR, which is also used in the bibliography. I would suggest introducing the acronym PNRR after the English name in line 177, like this: “the National Recovery and Resilience Plan (PNRR)”. This would help a non-Italian reader to understand what is what. Note also that reference [49] has a different font/typesetting.
Paradata about the acquisitions are given discursively but extensively. I am confident that in the work reported here they are recorded precisely in a structured way, which would be boring to report in the paper, so the discursive description used by the authors is indeed more than satisfactory. Moreover, as far as I know there is no agreed/standardized way of recording the acquisition paradata. Thus no change to the text is required.
Finally, I take the opportunity of this review to invite the authors, well-known researchers of international value, to participate in a just started academic discussion about the concept of Digital Twin. They identify this as the 3D model(s), a choice that is widely accepted by researchers in 3D modeling (see for example the work on the Notre Dame restoration) but is questioned by others. I musu admit that this identification is accepted by 99% of the researchers and argued only by 1%. It seems to me that some interesting considerations sparse in this paper, for example about the intangible values of the Appian way and the presence of a multitude of 3D and geographic models enriched by additional documentation, as described in the paper, support questioning the mainstream and considering an alternate concept of Digital Twin. I repeat that these considerations do not affect this paper, which is of outstanding value and a significant contribution to the field.
Author Response
Dear reviewer,
We sincerely thank you for your invaluable guidance and efforts in reviewing our work. Your insightful feedback has contributed significantly to enhancing the quality and clarity of our research. As a result of diligently following your recommendations, we have made the following improvements to our work – we highlight our replies to the comments in blue (see the attached word file). The major revision have been added to paragraph 8.
This is an excellent paper and deserves publication as is.
I have only a very very minor suggestion. On line 177, authors mention “the National Recovery and Resilience Plan” and in line 178 they cite the same with the Italian acronym PNRR, which is also used in the bibliography. I would suggest introducing the acronym PNRR after the English name in line 177, like this: “the National Recovery and Resilience Plan (PNRR)”. This would help a non-Italian reader to understand what is what. Note also that reference [49] has a different font/typesetting.
The reference has been updated.
Paradata about the acquisitions are given discursively but extensively. I am confident that in the work reported here they are recorded precisely in a structured way, which would be boring to report in the paper, so the discursive description used by the authors is indeed more than satisfactory. Moreover, as far as I know there is no agreed/standardized way of recording the acquisition paradata. Thus no change to the text is required.
It is correct: we have underlined that the paradata about acquisitions are structured according to the PAAA (MIC-Ministero della Cultura) Cultural Heritage data structure, inheriting also the vocabulary and semantics implemented in past experiences (as an Vault Vocabulary for HBIM and GIS across EU) [54] and on 3D Quality Models managing the complexity [62]. [pp 54 and 40].
Finally, I take the opportunity of this review to invite the authors, well-known researchers of international value, to participate in a just started academic discussion about the concept of Digital Twin. They identify this as the 3D model(s), a choice that is widely accepted by researchers in 3D modeling (see for example the work on the Notre Dame restoration) but is questioned by others. I musu admit that this identification is accepted by 99% of the researchers and argued only by 1%. It seems to me that some interesting considerations sparse in this paper, for example about the intangible values of the Appian way and the presence of a multitude of 3D and geographic models enriched by additional documentation, as described in the paper, support questioning the mainstream and considering an alternate concept of Digital Twin. I repeat that these considerations do not affect this paper, which is of outstanding value and a significant contribution to the field.
We’d like to thank you for your input on sharing academic discussion on DT we agreed on. And we explicitly integrated this concept [53] in the sub-paragraph 8.2 within the par.8 we completely revised as upped described [pp.34]
